

# Technical Note: Flagging inconsistencies in flux tower data

Martin Jung[1], Jacob Nelson[1], Mirco Migliavacca[2], Tarek El-Madany[1], Dario Papale[3,4], Markus Reichstein[1], Sophia Walther[1], Thomas Wutzler[1]

[1] Department of Biogeochemical Integration, Max Planck Institute for Biogeochemistry, Jena, 07745, Germany
[2] European Commission, Joint Research Centre, Ispra, Italy
[3] University of Tuscia DIBAF, Viterbo, 01100, Italy
[4] EuroMediterranean Center on Climate Change CMCC IAFES, Viterbo, 01100, Italy

*Correspondence to*: Martin Jung (mjung@bgc-jena.mpg.de)

**Abstract.** Global collections of synthesized flux tower data such as FLUXNET have accelerated scientific progress beyond the eddy covariance community. However, remaining data issues in FLUXNET data pose challenges for users, particularly for multi-site synthesis and modeling activities.

Here we present complementary consistency flags (C2F) for flux tower data, which rely on multiple indications of inconsistency among variables, along with a methodology to detect discontinuities in time series. The C2F relates to carbon
and energy fluxes as well as to core meteorological variables, and consists of: (1) flags for daily data values, (2) flags for entire site variables, (3) flags at time stamps that mark large discontinuities in the time series. The flagging is primarily based on combining outlier scores from a set of predefined relationships among variables. The methodology to detect break points in the time series is based on a non-parametric test for the difference of distributions of model residuals.

Applying C2F to the FLUXNET 2015 dataset reveals that: (1) Among the considered variables, gross primary productivity
and ecosystem respiration data were flagged most frequently, in particular during rain pulses under dry and hot conditions. This information is useful for modelling and analysing ecohydrological responses. (2) there are elevated flagging frequencies for radiation variables (shortwave, photosynthetically active, and net). This information can improve the interpretation and modelling of ecosystem fluxes with respect to issues in the driver. (3) The majority of long-term sites show temporal discontinuities in the time series of latent energy, net ecosystem exchange, and radiation variables. This should be useful for
carefully assessing the results on interannual variations and trends of ecosystem fluxes.

The C2F methodology is flexible for customizing, and allows for varying the desired strictness of consistency. We discuss the limitations of the approach that can present starting points for future improvements.

## 1 Introduction

The eddy covariance (EC) technique is widely used to assess the carbon dioxide ($CO_2$), water, energy, and other GHGs fluxes
between the surface and the atmosphere. Employed across major biomes globally, it counts thousands of stations distributed in all continents and often organized in regional networks (Baldocchi, 2020). Then, the FLUXNET initiative organized global



data collections and synthesis datasets such as the Marconi collection (Falge et al., 2005), the LaThuile dataset (Baldocchi et al., 2009), and the FLUXNET2015 (Pastorello et al., 2020), which have become a backbone for global ecosystem research (Baldocchi, 2020).

Flux tower measurements and associated data processing are complex and often subject to site-specific conceptual, technical, and logistic challenges. Principal investigators (PIs) of EC sites voluntarily provide their data to Regional Networks or directly to FLUXNET under a common data policy and standard format. The data include half-hourly or hourly biometeorological, environmental, and fluxes variables, all calculated and averaged by the PIs from the high-frequency raw meteorological and EC data. Before submission to the networks, the PIs generally apply a set of quality checks and quality assessment (QA/QC)

procedures (Metzger et al., 2012; Vitale et al., 2020) and site specific data filters as well as spectral corrections. This processing applied by the single groups is not strongly standardized. Thus, there is a high level of heterogeneity among sites concerning the completeness and effectiveness of applied quality control routines, detailed meta-data of instrumentation and applied processing as well as with respect to the availability of measured and reported variables.

The FLUXNET community developed a series of standardized tools for 1) reviewing critical metadata for the processing (e.g.,

site identifier, coordinates, reported time zone, and instrumentation height),  2) flagging meteorological data of questionable quality based on semi-automated visual checks of the relationships among different radiation variables (Pastorello et al., 2014), 3) filtering fluxes collected in low turbulence periods where the assumptions of the technique are not met (the so-called u* filtering, (Papale et al., 2006)), 4) gap-filling of missing data (Reichstein et al., 2005; Papale et al., 2006), and 5) partitioning of NEE into RECO and GPP components (Reichstein et al., 2005; Lasslop et al., 2010), and uncertainty calculations (Pastorello

et al., 2020). These tools are also organized in a set of routines (ONEFlux – https://github.com/Fluxnet/oneflux) that have been used in the FLUXNET2015 collection and continental networks releases (e.g., AmeriFlux FLUXNET product, ICOS Level2 data, Drought2018) and WarmWinter2020 collections). The routinely provided QC information for flux tower data informs primarily about the presence of an accepted measurement and the degree and quality of the gap-filling estimate, while potential issues in the underlying measurements may not be indicated.

Despite the effort to continuously develop and update standardized and common post-processing routines for FLUXNET, some measurement issues and inconsistencies between variables cannot be easily detected and flagged automatically - data quality relies also on the initial procedures applied by the PIs. This includes potential discontinuities in the time series due to undocumented changes in instrumentation or processing, which have developed over the last years and decades. To reduce the effect of differences in data treatment and QC between sites, some of the more structured networks, such ICOS in Europe

(Franz et al., 2018; Heiskanen et al., 2022) and NEON in the USA (Schimel et al., 2007) started to standardize the setup and methods (Franz et al., 2018; Rebmann et al., 2018) and the processing (Sabbatini et al., 2018) according to strict protocols, together with the collection of full and detailed metadata. This facilitates centralized processing from the raw data and reprocessing with more advanced methods as they become available (Vitale et al., 2020), taking into consideration all the changes in  the measurement setup and ecosystem state. Developing standardized processing and QC that works robustly and

reliably for all cases is very challenging as ecosystems, land surface, and (micro-) meteorological conditions can be very





heterogeneous between sites. Thus, the standardized methods used are not perfect and site specific issues can persist. For example, the night-time based Net ecosystem exchange (NEE) partitioning method (Reichstein et al., 2005) might give unreliable Gross Primary Productivity (GPP) and Ecosystem respiration (Reco) results when temperature is not the main driver of respiration.

The remaining issues and inconsistencies in FLUXNET data pose limitations for synthesis studies, particularly for process-based or machine learning based model calibration and evaluation. The degree to which model-data mismatches are due to model deficiencies or perhaps data issues, either in the fluxes or in the meteorological data used as model input, is typically hard to judge, especially by non-EC experts. This can limit progress in improving the modelling for certain aspects. From the machine learning based modelling perspective, some unanswered example questions on the contribution of potential flux tower

data issues include: (1) Can we predict the interannual variability of sensible heat flux much better than that of latent heat flux due to differential observational uncertainties? (2) To what extent is the low skill in predicting NEE interannual variability at FLUXNET site level due to temporal discontinuities arising from changes in instrumentation and set-up. (3) How much of the issue to model drought effects in GPP is due to flux partitioning problems? (4) Where is the optimal trade-off between data quantity and data quality used for training machine learning models? To progress on such questions, we need a complementary

data consistency control applicable across the network's heterogeneous data conditions and core flux tower variables, following objective principles and allowing for varying the strictness of tolerated inconsistency.

Here we address this challenge of providing complementary consistency flags (C2F) for FLUXNET data. It complements the quality control applied by PIs and ONEFLUX as it is exclusively based on inconsistencies among measured variables. The degree of allowed inconsistency, a strictness parameter, has an interpretable basis and can be varied by the user. The underlying

framework allows for extending and customizing the methodology as better knowledge or experience becomes available. Its objective principles facilitate full automatization and thus integration into processing pipelines for e.g. FLUXCOM or ONEFLUX. It delivers: (1) Flags for daily data points, as well as flags for entire site-variables for ecosystem fluxes and core meteorological variables, and (2) times at which large discontinuities in the data occur and may indicate issues due to changes in the instrumentation, setup or footprint.

We describe the C2F methodology in section (2) along with some examples for illustration – for clarity of the overall principle, methodological details are bundled in the last subsection. In section (3) we synthesize results of C2F for the FLUXNET 2015 dataset, where we also investigate whether flagging patterns are systematic. The discussion in section (4) focusses on methodological considerations, recommendations for potential applications, and interpretation.

## 2 Materials and methods

### 2.1 FLUXNET Dataset

The FLUXNET2015 Dataset (Pastorello et al., 2020) is a collection of meteorological and flux data measured at 212 sites and collected from multiple regional flux networks. The geographical location of the sites ranges from a latitude of 37.5 S to 79



N and covers all the main plant functional types. Compared to previous releases of flux observations, the FLUXNET2015 Dataset includes several improvements, in particular to the data quality control protocols and the data processing pipeline 100 (Pastorello et al., 2020).

The complementary data consistency checks described here were developed and applied to daily data (temporal average) for the variables mentioned in table 1. At the beginning we filtered out values with fqcOK < 0.8, i.e. retaining only data points that are based on at least 80% of measured data or gap-filled with high confidence. The C2F were then applied to only those data points.


| Acronym | Name |
|---|---|
| GPP_NT | Gross primary productivity estimated based on the night-time flux partitioning method (Reichstein et al. 2005) |
| GPP_DT | Gross primary productivity estimated based on the day-time flux partitioning method (Lasslop et al. 2010) |
| RECO_NT | Ecosystem respiration estimated based on the night-time flux partitioning method (Reichstein et al. 2005) |
| RECO_DT | Ecosystem respiration estimated based on the day-time flux partitioning method (Lasslop et al. 2010) |
| NEE | Net ecosystem exchange |
| LE | Latent energy |
| H | Sensible heat |
| NETRAD | Net radiation |
| SW_IN | Shortwave incoming radiation |
| PPFD_IN | Photosynthetic Photon Flux Density |
| TA | Air temperature |
| VPD | Vapour pressure deficit |

**Table 1: List of variables for which complementary consistency flags are derived.**

**2.2 Flagging inconsistencies among variables**

The approach described here is based primarily on multiple indications of inconsistency between variables for a given site. Its final output is a boolean flag for every daily data point and target variable listed in Table 1, where TRUE indicates an 110 inconsistency. Additionally, it reports a boolean flag for entire site variables, which is based mainly on between-site inconsistencies of relationships.

C2F is rooted in defining consistency constraints among variables. A constraint refers to an expected relationship of a target variable with other variables, which is used to calculate an outlier score for every data point (section 2.2.1). The flagging





procedure is then based on three main consecutive steps: (1) calculating the outlier score for each pre-defined constraint
(section 2.2.2), (2) for each target variable, the associated outlier scores are combined to yield an inconsistency score, which
considers multiple indications of inconsistency (section 2.2.3), and (3) flagging is then based on thresholding the inconsistency
score along with taking further considerations into account (section 2.2.4 and 2.2.5).

### 2.2.1 Constraints

We define relationships among variables where unusually large deviations from the relationship, i.e. outlier points, indicate a
potential issue in at least one of the variables. For example, we expect a strong relationship between PPFD_IN and SW_IN.
We refer to such defined relationships as "constraints" in the following. The constraints are based on expert knowledge with
some empirical and theoretical justification (Table 2), while we make a qualitative distinction between hard and soft constraints
during flagging (see section 2.2.4) according to their conceptual strength in attributing outliers to data issues. For example,
outliers in the NETRAD vs SW_IN relationship are not necessarily related to data issues but could originate from sudden
changes in the albedo, e.g. due to snow, harvest or disturbance. Broadly speaking, hard constraints refer to physical
relationships between variables where large inconsistency among them reflect a problem in the data with high confidence. Soft
constraints refer to relationships based on an underlying model with some assumptions and uncertainty. For some of the soft
constraints we know that for certain conditions the relationship breaks due to violations of assumptions (e.g. NETRAD vs
SW_IN for negative net radiation) such that we can mask those out from the beginning. We use primarily bivariate constraints,
i.e. linear relationships between two variables, and machine learning constraints, i.e. where a target variable is modelled from
a set of predictor variables. Table 2 lists and describes the constraints defined here – more details are available in section 2.4.




| Constraint | Rationale | Median correlation |
|---|---|---|
| SW_IN vs PPFD_IN | Because this constraint is associated with a very tight physical link and empirical relationship it is classified as hard constraint. | 0.998 |
| NETRAD vs SW_IN | Incoming solar radiation dominates the temporal variations of net radiation on a daily resolution. Because net radiation includes outgoing and longwave components that depend on other factors these constraints are classified as soft constraints. When solar radiation is low, e.g. in winter time conditions, net radiation can become negative and the relationship meaningless such that these constraints are only evaluated for data points with positive net radiation. | 0.955 |
| NETRAD vs PPFD_IN | | 0.961 |
| NETRAD vs LE+H | Net radiation is linked to the sum of latent and sensible heat fluxes through the energy balance. Contributions by storage changes are neglected for simplicity – their effect is small on daily resolution and corresponding data are not always available. The empirical relationship is typically not on the 1:1 line due to the pervasive energy balance closure gap problem. Due to the physical nature of the relationship, we classified it as hard constraint. | 0.955 |
| GPP_NT vs GPP_DT | As these pairs are relationships between alternatives from different methods of the same quantity they are classified as hard constraints. | 0.966 |
| TER_NT vs TER_DT | | 0.882 |
| RECO_NT_NIGHT vs NEE_NIGHT | | 0.950 |
| RECO_DT_NIGHT vs NEE_NIGHT | | 0.838 |
| GPP_NT*sqrt(VPD) vs LE | This reflects a water use efficiency model based on optimality considerations for stomatal conductance, which had been tested with EC data. To reduce confounding effects by elevated evaporation, the constraint is not evaluated for rain days. Due to the assumptions of the model we classified it as soft constraint. For better independence among constraints for the same target variable, only the minimum of the outlier scores from both variants is assigned to LE. | 0.888 |
| GPP_DT*sqrt(VPD) vs LE | | 0.880 |





| NEE ustar uncertainty | Uncertainties due to Friction velocity ($u_*$) filtering that are unusual high can point to violations of assumptions underlying NEE measurements by EC. Because $u_*$ uncertainty estimates depend on gapfilling methods it is classified as soft constraint. | n.a. |
|---|---|---|
| TA vs TA ERA-5 | Because site to pixel relationships with ERA5 meteorological reanalysis can be affected by uncertainty of ERA5 and footprint mismatches they are treated as soft constraints. They were included due to lack of constraints for TA and VPD from tower measurements only. | 0.991 |
| VPD vs VPD ERA-5 | | 0.911 |
| Machine learning | Predicting a flux tower variable based on other flux tower variables includes uncertainties e.g. due to missing predictors or quality issues in the predictors. Therefore, these are classified as soft constraints. Variable specific predictor sets were chosen to increase independence among constraints (see section 2.4.2). | 0.933 – 0.992 |

**Table 2: Rationale for selected constraints. Median correlation refers to the median Pearson correlation coefficient calculated for each site when executing the constraints, i.e. based on daily data and with outliers removed (see section 2.4 for details).**

Each constraint is assigned to one or several target variables involved in the relationship. In the SW_IN vs PPFD_IN example, the constraint is assigned to PPFD_IN and SW_IN because both are equally likely to be right or wrong for an outlier point. Likewise, the constraint LE+H vs NETRAD is assigned to LE, H, and NETRAD. The constraint RECO_NIGHT vs NEE_NIGHT is only assigned to RECO_NIGHT since it indicates primarily issues of the underlying flux-partitioning model, rather than issues of measured NEE at night. Table 3 summarizes the assignments of constraints to target variables used here,

while the methodology is flexible in adding, removing, or modifying constraints.





| Constraints | GPP_NT | GPP_DT | RECO_NT | RECO_DT | NEE | LE | H | NETRAD | SW_IN | PPFD_IN | TA | VPD |
|---|---|---|---|---|---|---|---|---|---|---|---|---|
| **Machine learning** | S | S | S | S | S | S | S | S | S | S | S | S |
| **u\*** | | | | | H | | | | | | | |
| **GPP_NT vs GPP_DT** | H | H | | | | | | | | | | |
| **[1]GPP_NT\*sqrt(VPD) vs LE** | S | | | | | S | | | | | | S |
| **[1]GPP_DT\*sqrt(VPD) vs LE** | | S | | | | | | | | | | |
| **RECO_NT vs RECO_DT** | | | H | H | | | | | | | | |
| **RECO_NT_NIGHT vs NEE_NIGHT** | | | H | | | | | | | | | |
| **RECO_DT_NIGHT vs NEE_NIGHT** | | | | H | | | | | | | | |
| **NETRAD vs LE+H** | | | | | | H | H | H | | | | |
| **[2]NETRAD vs SW_IN** | | | | | | | | S | S | | | |
| **[2]NETRAD vs PPFD_IN** | | | | | | | | S | | S | | |
| **SW_IN vs PPFD_IN** | | | | | | | | | H | H | | |
| **TA vs TA ERA-5** | | | | | | | | | | | S | |
| **VPD vs VPD ERA-5** | | | | | | | | | | | | S |
| **# constraints for samples (within site)** | 3 | 3 | 3 | 3 | 2 | 3 | 2 | 4 | 3 | 3 | 2 | 3 |
| **# constraints for variable (between sites)** | 4 | 4 | 4 | 4 | 3 | 4 | 3 | 5 | 4 | 4 | 3 | 4 |

**Table 3: List of hard (H) and soft (S) constraints.** [1] excluding rain days. [2] excluding NETRAD < 0.





### 2.2.2 Outlier score for data points

The execution of each constraint delivers an outlier score eventually (Fig. 1) for each single data point i which quantifies its
anomalous deviation from the expected conceptual relationship. The calculation considers all daily data points available for a
site in contrast to processing by year or moving windows. The outlier score is based on the residuals of the relationship for
most constraints ($R_i = Y^{obs}_i - Y^{pred}_i$). Instead of categorizing outliers from the beginning we calculate a continuous outlier score
corresponding to the widely used 'boxplot rule', while we take heteroscedasticity and asymmetry of distributions into account
(see section 2.4.1). The boxplot rule labels data points as outliers if they are 1.5 units of interquartile range (IQR) apart from
the first or third quartiles; far outliers are often labeled when the distance exceeds 3 times IQR. How many units of IQR (nIQR)
should be chosen can be subjective and application dependent – it is essentially a strictness parameter that one might want to
vary approximately between 1.5 and 3, corresponding to a gradient of 'strict' consistency retaining less data to 'loose'
consistency retaining more data. Thus, the parameter nIQR is the key consistency strictness parameter of our approach with a
default value of 3 that can be varied by the user. Through this definition of the outlier scores,  outlier scores from different
constraints are independent of units, comparable, and therefore combinable among different constraints, which is an important
prerequisite of our approach to calculate the inconsistency score. Implementation details for the different constraint types are
available in section 2.4.

Figure 1 illustrates the overall concept of the outlier score for the example of a machine learning constraint of GPP_NT for a
dry site in the United States. The machine learning model was trained with meteorological predictors and captures the flux
patterns very well in general (Fig. 1 top). However, it does for example not predict larger negative GPP values that are present
in the F15 data. The residuals show a clear pattern of heteroscedasticity, i.e. residuals tend to be larger when GPP is larger (Fig
1 middle). By taking this heteroscedasticity into account we can identify the large negative GPP values in F15 as outliers (Fig.
1 bottom). This is because the residuals are large relative to the expected narrow distribution of residuals for small GPP (see
section 2.4.1 for more details).

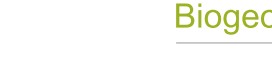



**Figure 1: Illustration of the derivation of the outlier score for a constraint. This example is for the machine learning constraint for GPP_NT for US-Wkg. Observed and predicted values (top) are used to calculate residuals and how the distribution of residuals varies with the predicted value (middle). The outlier score (bottom) measures the distance of the residuals to the quartiles in units of interquartile range (nIQR).**

**2.2.3 Inconsistency score of a data point**

We first calculate the respective outlier scores for all constraints and then combine them into a inconsistency score for each target variable (Table 3). Thus, the inconsistency score takes multiple indications of inconsistency into account. In most cases, the inconsistency score is the second largest outlier score from all available constraints for a target variable and data point, normalized by the specified strictness parameter nIQR (see section 2.4.5 for details). Conceptually, this refers to the situation



where at least two constraints need to identify a data point as an outlier. The inconsistency score is undefined if less than two outlier scores are available, while flagging is still possible if a hard constraint indicates an outlier (see section 2.2.4).

The procedure of calculating the inconsistency score is advantageous in the presence of missing data in specific outlier scores and therefore acknowledges the heterogeneity in terms of availability of variables. Looking at more than one soft constraint also addresses the robustness problem - if there is a real problem in the data, it should be evident from several constraints.

Individual soft constraints may indicate an outlier due to violations of underlying assumptions (i.e. false positives), while false positive outlier indication for the same data point by different soft constraints is very unlikely if the constraints are independent. Since the inconsistency score pools the outlier scores of different constraints it also addresses the variable attribution problem that we would have for most bivariate constraints when looking at a single constraint only. For example, we cannot attribute an outlier indicated in the PPFD_IN vs SW_IN constraint to an issue in either of the variables.  Instead, the inconsistency

scores for the two variables consider all available constraints and provide an indication which variable shows a data issue. We will illustrate this by an example of deriving the inconsistency score of SW_IN for a site in France (Fig.2). Figure 2 shows the three constraints available for SW_IN where the respective outlier scores scale with the color: the bivariate relationships with PPFD_IN and with NETRAD respectively, as well as the machine learning based constraint for SW_IN. In the scatter plots, we see two major patterns of inconsistency: (1) SW_IN scales differently with PPFD_IN for a subset of the data, and (2) all

three constraints indicate an issue related to some values of zero for SW_IN. In the time series plots for SW_IN, we see that the first pattern of inconsistency between SW_IN and PPFD_IN occurs for a long consecutive period in 1997, and that the second pattern occurs in 2002 where SW_IN is constant at zero. The inconsistency score for SW_IN shows the latter issue accordingly since it is present in multiple constraints, while it shows no major issue for SW_IN in 1997 where there was the inconsistency with PPFD_IN only.






Fig. 2: Illustration of the derivation of the inconsistency score for a variable, here SW_IN for the site FR-LBr, based on outlier scores of different constraints.The color scales with the outlier or inconsistency score in the same way across panels.





### 2.2.4 Flagging data points

The first step of flagging data points for a target variable is based on thresholding the corresponding inconsistency score at >1. Please note that this corresponds to the set nIQR threshold, which was used to normalize the outlier scores for the computation of the inconsistency score (section 2.2.3). In the second step, we iterate over the following procedures:

(1) We propagate flagged data points to dependent variables (e.g. from SW_IN to GPP_DT since the latter depends on the former, see table 4 for considered dependencies).

(2) If two flagged data points are less than a few days apart (default = 4 days), we additionally flag these data points in between. This is done because data issues often appear in sequence e.g. due to instrumentation issues or moving window processing of the flux partitioning, while the inconsistency score may not always exceed 1.

(3) If hard constraints indicated an outlier data point but none of the target variables assigned to this constraint were flagged yet, we force flagging based on an attribution scheme (see section 2.4.6). Forcing flagging for hard constraints

(e.g. PPFD_IN vs SW_IN) is done because we assume that there must be a data issue in at least one variable involved if an outlier is identified by this constraint, according to our definition of hard constraints.

| |
|---|
| NEE → GPP_NT, GPP_DT, RECO_NT, RECO_DT |
| GPP_NT ←→ RECO_NT |
| GPP_DT ←→ RECO_DT |
| TA → GPP_NT, GPP_DT, RECO_NT, RECO_DT |
| SW_IN → GPP_DT, RECO_DT |
| VPD → GPP_DT, RECO_DT |

**Table 4: Predefined dependencies for flag propagation. Arrows indicate the direction of flag propagation.**

Figure 3 illustrates the overall approach for flagging data points for SW_IN using some values as examples that are not necessarily typical but hopefully educational. PPFD_IN is included for necessary context of the hard constraint defined

between them. The iterative procedure associated with the forcing of hard constraints (see section 2.4.6 for details) is omitted from the visualization for clarity. The principle of forcing flagging for outliers of hard constraints is illustrated by orange colours. In this example the hard constraint suggests two outliers (the first two data points) that were initially not flagged by either SW_IN or PPFD_IN based on thresholding the inconsistency score. The attribution scheme then allocates the first data point to an issue of PPFD_IN and the second data point to SW_IN based on the constellations of their inconsistency scores

(the section2.4.6 for details).





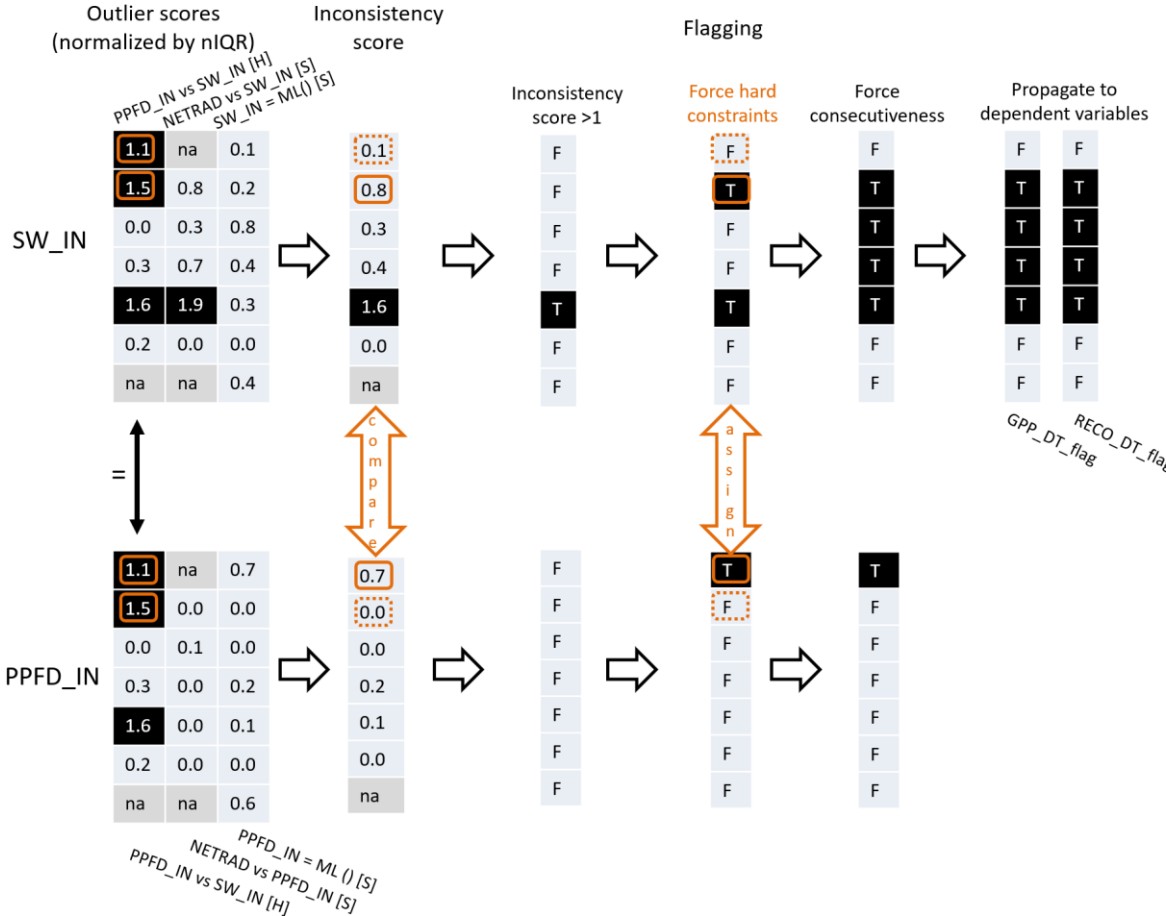

**Figure 3: Schematic flow chart for flagging data points for the example SW_IN. Data points with black background show values above the threshold nIQR or flagging (T=True, F=False). na indicates a missing value. The first two data points show outliers for the PPFD_IN vs SW_IN hard constraint, where PPFD is flagged for the first data point, while for the second data point, SW_IN is flagged in the process of forcing hard constraints (highlighted by orange colours) by considering the inconsistency scores for SW_IN and PPFD_IN. For the fifth data point, SW_IN is flagged because the inconsistency score exceeds nIQR. Forcing consecutiveness additionally rejects the third and fourth data point for SW_IN. Flagged SW_IN data points are propagated to the daytime flux partitioning variables due to dependence on SW_IN.**

Turing back to our previous SW_IN example for the site in France, we see that the flagging has correctly flagged the PPFD_IN values in 1997, and flagged the SW_IN values in 2002 (Figure 4). The SW_IN flag is propagated to GPP_DT which shows the same problem as SW_IN in 2002.







**Figure 4: Derived flags for SW_IN, PPFD_IN, NETRAD, and GPP_DT for FR-LBr for a strict (nIQR=1.5) and a loose (nIQR=3)**
**consistency setting.**





To further illustrate the flagging behavior, we look again at the dry site from the US (Fig. 5). The flagged data points due to the inconsistency score (red points) are dominated by negative GPP values for GPP_NT. Many of those also correspond to outliers of the GPP_NT vs GPP_DT constraint (blue stripes), while the attribution has also rejected GPP_DT sometimes. For RECO_NT we see that flagged values are dominated by the hard constraints outliers of the relationship between nighttime

RECO_NT with nighttime NEE (magenta stripes). These data points occur predominately in the dry season indicating issues of the night-time based flux partitioning method ((Reichstein et al., 2005)). These data points, often associated with negative GPP_NT and elevated NEE, are often rejected independently for GPP_NT based on the inconsistency score. The propagation ensures that flags are finally consistent and identical for GPP_NT and RECO_NT, as well for GPP_DT and RECO_DT respectively.






**Figure 5: Illustration for flagging GPP and Reco values from the nighttime and the daytime partitioning method for US-Wkg. The flagging is based on default values (nIQR=3).**





### 2.2.5 Flagging entire variables

So far, we have aimed at flagging inconsistent data points for a target variable within a site, given the information available for that site. Now we aim at identifying if an entire site-variable time series, measured at one site, behaves unusually and should perhaps be flagged. If for example a variable would be in a wrong unit, the regression approaches used in the within-site outlier detection will not catch it while e.g. the slope of the regression will appear as unusual compared to the distribution of slopes from the same constraint available across sites. In a similar notion, if the relationship between two variables for a

constraint is unusually weak for a site, the within site processing will not catch this because it looks only for outliers, given the distribution of residuals within the site.

  The constraints for the between site entire variables flagging are based on: (1) the performance of the relationship from the machine learning constraints, (2) parameters of the linear model of the bivariate constraints in combination with its performance (see Figure 6 for an illustration), (3) the fraction of flagged values for a variable from the within-site processing,

(4) a metric related to the NEE u*-uncertainty per site as diagnosed by the within-site processing (see section 2.4.4). Because we add the fraction of flagged values as between site-variable constraints, the number of between-site constraints per variable is one more than for the within site processing (see Table 3). The above mentioned diagnostics are converted into outlier scores considering the distribution across sites (see section 2.4.7 for details). The calculation of the inconsistency score per site-variable and following flagging of site-variables follows the methodology described for the within-site processing except that

we do not apply the consecutiveness rule in the flagging procedure as it is meaningless here.

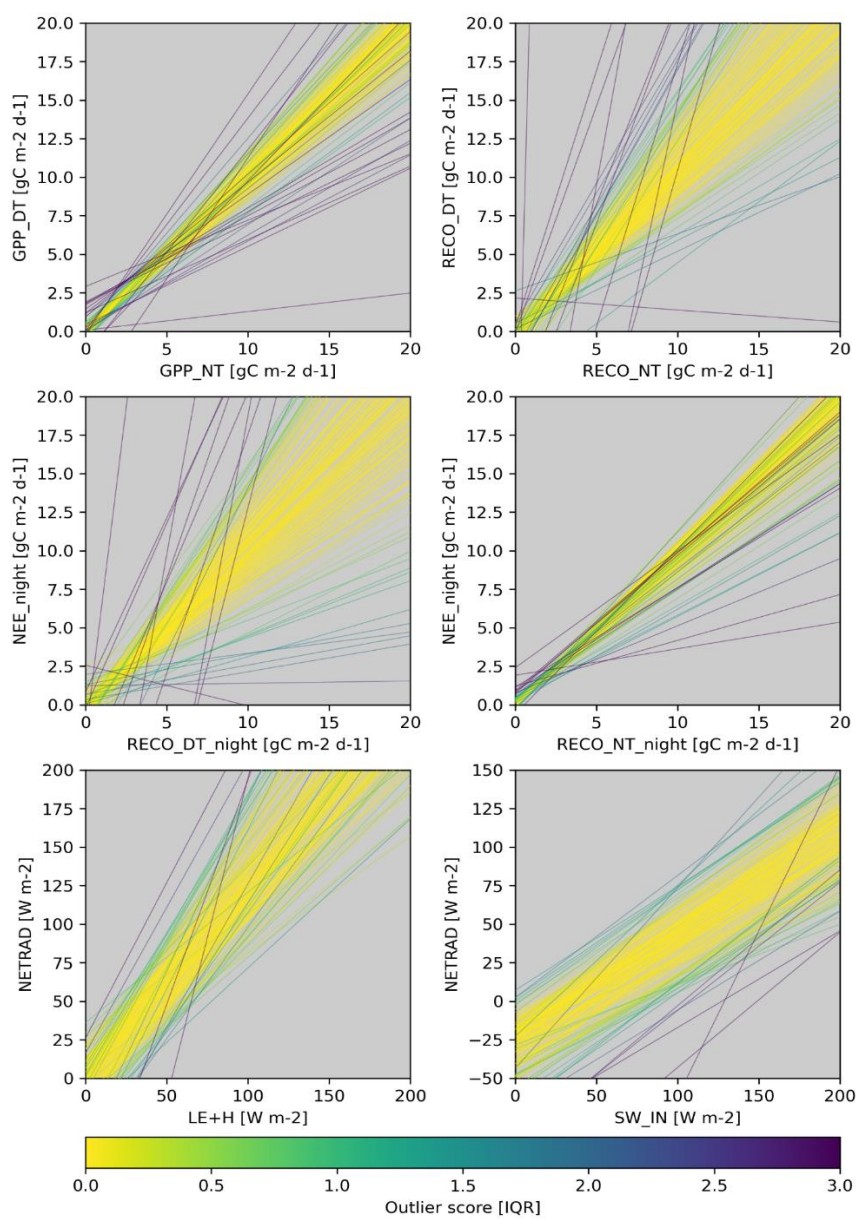

**Figure 6: Estimated regression lines for each site for some bivariate constraints. Each line corresponds to a site, colors correspond to the derived outlier score for the specific constraint.**

**2.3 Identifying temporal discontinuities**

Here we aim at identifying systematic changes in the distribution of flux tower data within a time series that could point to data artefacts, e.g. due to instrumentation or setup changes. The basic principle is to move from the beginning of a time series to the end. At each time step we assess the difference in the distribution between the data before the current time step and the





distribution after the current time step (see section 2.4.8 for details). This yields a new time series of the test statistic for the

difference of distributions for which we seek the maximum (Fig.7). We use a non-parametric test for the equality of two

distributions based on their energy distance (Szekely and Rizzo, 2004) – intuitively energy distance can be understood as the

amount of work necessary to transform one distribution into the other.

The change in distribution is assessed based on residuals of two machine learning models for the target variable and site. The

first model uses meteorological conditions as input, the second model uses only seasonal information as input. The residuals

of both models are normalized to account for heteroscedasticity. The test statistic for the difference in distribution is calculated

based on distances in two-dimensional space, where the two dimensions correspond to the time series of the normalized

residuals of the two models. The rationale for this approach is discussed in section 4.1.2.

The breakpoint detection was setup as a recursive partitioning where the time series is iteratively split into segments. For

example, we first run the breakpoint detection on the full time series. Then the time series is split into two segments according

to where we found the largest difference in distributions. Then the breakpoint detection is run for both segments  again. This

procedure continues until not sufficient data are in the segments (default = 100 data points). For every split we calculate and

store a break severity metric that is used to calculate a corresponding outlier score (see section 2.4.8).

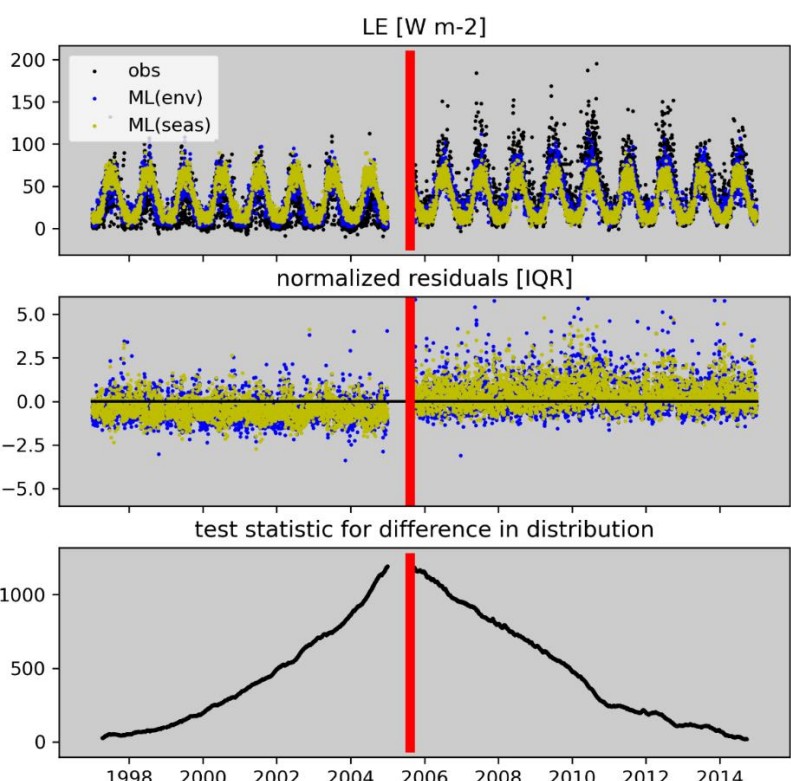

**Figure 7: Illustration of the break point detection for CH-Dav. The top panel shows the observed time series of daily LE and model**
**predictions. The middle panel shows the normalized residuals that are passed to the breakpoint detection. The bottom panel shows**
**the estimated test statistic for the difference in distributions, which is maximized at the red bar, which denotes the detected break.**





## 2.4 Methodological details

### 2.4.1 Calculation of outlier score for data points

The outlier score for data points derived for a specific constraint and site is calculated based on the residuals ($R_i = Y^{obs}_i -$
$Y^{pred}_i$). The predictions come from a linear regression model for bivariate constraints and from a Random Forest model for machine learning based constraints. We take heteroscedasticity into account, i.e. that the distribution of residuals varies with magnitude, to avoid systematic biases of outlier detection with magnitude (Figure 8). To do so, we estimate how the $25^{th}$, $50^{th}$ and $75^{th}$ percentiles of the residuals vary for each data point i, depending on the magnitude of $Y^{pred}$ (see below). Our outlier score O measures the maximum distance to the first and third quartiles of the residuals in units of IQR, while we also take
potential asymmetry in the distribution of residuals into account:

$$O_i = max\left(\frac{R_i - P75_i}{2(P75_i - P50_i)}, \frac{P25_i - R_i}{2(P50_i - P25_i)}\right) \qquad (1)$$

The denominators refer to the interquartile range of the distribution of R represented by two half distributions, one for each side (Schwertman et al., 2004) estimated by 2 times the distance between median and the respective first or third quartile.

We developed a non-parametric method to estimate the heteroscedastic behaviour of residuals that is capable of handling
heterogenous patterns of heteroscedasticity found for different constraints and sites, and the typically very skewed data distribution of $Y^{pred}$.

(1) We first calculate the residuals $R_i$ and sort them ascending according to $Y^{pred}$.
(2) We calculate P25 and P75 of R included in a moving (non-overlapping) step window (see Fig. 8 bottom right panel). Default window size is 100 data points. This is reduced if necessary to yield at least 15 estimates for P25
and P75 if the time series is short. This yields vectors of $Y^{pred}_w$ (mean), $P25_w$, and $P75_w$ which are typically shorter than the original R and $Y^{pred}$ vectors by a factor of 100. We observed that the estimated interquartile range ($iqr_w = P75_w - P25_w$) is occasionally (very close to or) zero, for example when flux partitioning results were truncated at 0 causing long consecutive periods of zero flux in winters for instance. This made the outlier score too sensitive to very small residuals even. To counteract this we imposed a minimum $iqr_w$, i.e. we take the $max(iqr_w, iqr_{min})$ and
adjust P75 and P25 accordingly if necessary. $iqr_{min}$ was chosen to be 7% of the standard deviation of $Y^{pred}$ – this heuristic choice is based on empirical trials and visual inspections.
(3) Since the step window approach does not provide an estimate for P25 and P75 for the smallest and largest values of $Y^{pred}$, linear regression is used to extrapolate $P25_w$ and $P75_w$ for the smallest and largest values in $Y^{pred}$. The linear regression uses the smallest or largest 10% of $Y^{pred}_w$ respectively, and corresponding $P25_w$, and $P75_w$ obtained from
step (2) and at least 5 data points. The linear regression then estimates $P25_w$ and $P75_w$ for the smallest and largest value of $Y^{pred}$ and those values are inserted in the respective vectors (see e.g. most right data point plotted in bottom right panel of Fig. 8).
(4) Since the empirical estimation of $P25_w$ and $P75_w$ is typically noisy we perform a lowess (locally weighted scatterplot smoothing, (Cleveland and Devlin, 1988)) filtering with a span of 20% of the length of the vector and by
also specifying $Y^{pred}_w$ as the corresponding exogenous variable locations. This yields a smooth variation of how $P25_w$ and $P75_w$ vary with $Y^{pred}_w$ (see plotted lines in bottom right panel of Fig. 8).
(5) The smoothed versions of $P25_w$ and $P75_w$ at locations $Y^{pred}_w$ are linearly interpolated at the locations of the original, full, $Y^{pred}$ vector to obtain the finally required $P25_i$ and $P75_i$.





The presence of outliers can inflate the estimated interquartile range of residuals and could result in false negatives, in
particular if the distribution of outliers is in some way systematic with the magnitude. To counteract this, steps (2) to (5)
are repeated several times and outliers detected in the current iteration (defined as exceeding nIQR=3) are masked out for
the calculations of the next iteration. The iteration stops when a stable set of outliers are found.

The procedure described above has been developed to obtain a reasonable solution with tractable computational costs since
the calculation of the heteroscedastic outlier score needs to be done for each site and constraint several times. Because the data
adaptive and non-parametric estimation of percentiles also accounts for systematic changes of the median residual with
predicted magnitude it effectively relaxes the linearity assumption of bivariate constraints. For some constraints this is an
advantage (e.g. for LE vs GPP*sqrt(VPD)), while for others it is a conceptual disadvantage (e.g. for NETRAD vs LE+H).

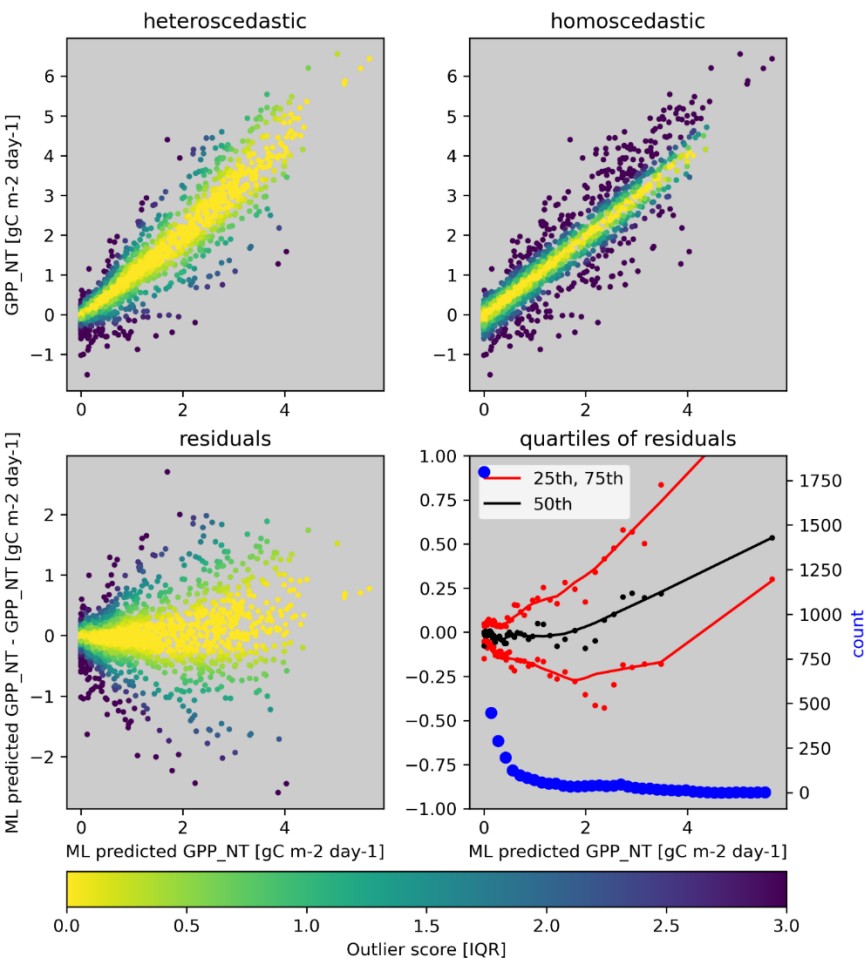

**Figure 8: Illustration of the behavior and calculation of the heteroscedastic outlier score for the machine learning constraint for**
**GPP_NT for US-Wkg (see also Fig. 1). Top row shows the effect of accounting for heteroscedastic residuals for the outlier score (on colour). Bottom left illustrates how the variability of residuals vary with magnitude. Bottom right shows the empirically estimated quartiles by the step window (dots) and the smoothed and interpolated estimates for each data point (lines). The blue dots show the frequency distribution of data points. The example corresponds to nIQR=3.**





**2.4.2 Implementation of machine learning constraints**

Inputs are a set of specified and gap-filled predictor variables, and the target variable Y of the constraint (not gap-filled). The predictor sets (Table 5) were chosen to increase independence among different constraints, e.g. that variables used in bivariate constraints for the same target variable are excluded from the predictor set (e.g. NETRAD and PPFD_IN are not used to model SW_IN with machine learning). The outputs are 1) the outlier score for each data point i which is used in the within-site processing, and 2) robust estimates of correlation and root mean squared error later used for flagging site-variables (see section 355 2.4.7). The algorithm is:

(1) Run a cross-validation with a machine learning algorithm to obtain the cross-validated predictions $Y^{pred}$. We chose a 3-fold cross-validation with random partitioning, and the scikit-learn implementation of Random Forests (Breiman, 2001) with default parameters as machine learning model.

(2) Calculate the outlier score $O^{ML}$ based on Y and $Y^{pred}$ cf eq 1.

(3) Calculate correlation and RMSE from Y and $Y^{pred}$ with outliers removed (i.e. where $O^{ML} < 3IQR$).

Steps (1)-(2) are iterated a few times where outliers identified in the current iteration are masked out in the training data for the next iteration.

Missing values in the predictor variables of the training data set were imputed using missForest (Stekhoven and Bühlmann, 2011), which is an iterative gap-filling procedure for a set of variables based on Random Forests. This was done to maximize 365 data availability and applicability. We calculated a set of water balance indicators from daily gap-filled precipitation (P) and evapotranspiration (E) to improve predictability for water-stressed conditions, especially because measured soil moisture contents are provided inconsistently. Those are added as predictors whenever measured soil moisture content was specified as predictor. These are cumulative water deficit (CWD), truncated cumulative water deficit (tCWD) with storage capacities of 15,50,100,150,200,250 mm, and detrended cumulative water balance (CWB):

$$CWD_t = min(CWD_{t-1}+P_t-E_t,0) \qquad\qquad (2)$$

$$tCWD_t^C = max(0,min(tCWD_{t-1}+P_t-E_t,C)) \qquad\qquad (3)$$

$$CWB_t = CWB_{t-1}+P_t-E_t \text{ (detrended)} \qquad\qquad (4)$$

where C is the specified storage capacity. Six variants of $tCWD_t^C$ with different C values of 15,50,100,150,200,250 mm were calculated.






| Target variable | Predictors |
|---|---|
| SW_IN | P, SW_IN_POT, LE, H, GPP_NT, RECO_NT, NEE, VPD, TA |
| PPFD_IN | P, SW_IN_POT, LE, H, GPP_NT, GPP_DT, RECO_NT, RECO_DT, NEE, VPD, TA |
| NETRAD | P, LW_OUT, SW_IN_POT, GPP_NT, GPP_DT, RECO_NT, RECO_DT, NEE, VPD, TA |
| TA | PPFD_IN, P, LW_OUT, SW_IN_POT, LE, H, NETRAD, NEE, SW_IN, SMI |
| VPD | PPFD_IN, P, LW_OUT, SW_IN_POT, LE, H, NETRAD, GPP_NT, RECO_NT, NEE, SW_IN, SMI |
| LE | PPFD_IN, P, LW_OUT, SW_IN_POT, VPD, TA, SW_IN, SMI |
| H | PPFD_IN, P, LW_OUT, SW_IN_POT, VPD, TA, SW_IN, SMI |
| NEE | PPFD_IN, P, LW_OUT, SW_IN_POT, NETRAD, VPD, TA, SW_IN, SMI |
| GPP_NT | PPFD_IN, P, LW_OUT, SW_IN_POT, NETRAD, VPD, SW_IN, SMI |
| GPP_DT | PPFD_IN, P, LW_OUT, SW_IN_POT, NETRAD, SMI |
| TER_NT | PPFD_IN, P, LW_OUT, SW_IN_POT, NETRAD, VPD, SW_IN, SMI |
| TER_DT | PPFD_IN, P, LW_OUT, SW_IN_POT, NETRAD, SMI |

**Table 5: List of predictor variables used for different target variables. SMI stands for soil moisture indicators and denote the set of measured (top and bottom) soil moisture (if available) and derived indicators $CWD_t$, $tCWD_t^C$, $CWB_t$**

### 2.4.3 Implementation of bivariate constraints

Inputs are two variables, Y1 and Y2, and the outputs are 1) an outlier score for each data point i which is used in the within-site processing, and 2) robust estimates of correlation, root mean squared error, slope and intercept of a linear model used later for flagging site-variables (see section 2.4.7). The algorithm is:

(1) Compute a robust linear regression with X=Y1 and Y=Y2. Predict $Y2^{pred}$ as a function of Y1 accordingly. Calculate the outlier score $O^{Y2,Y2pred}$ cf Eq 1.
(2) Repeat step (1) with X and Y swapped and obtain $O^{Y1,Y1pred}$
(3) Take the maximum to obtain the final outlier score for the bivariate constraints: $O^B = max(O^{Y1,Y1pred}, O^{Y2,Y2pred})$.
(4) Calculate an orthogonal regression between Y1 and Y2 but with outliers removed (i.e. where $O^B < 3IQR$) and report correlation, RMSE, slope and intercept.

As robust linear regression we used RANSAC (random sample consensus, (Fischler and Bolles, 1981)) implemented in
scikit-learn which is based on finding consensus among many linear models fitted to different random subsets of the data. We chose a subset of random 50% of the data and a maximum iteration limit of 200.





### 2.4.4 Implementation of u∗ uncertainty constraint for NEE

Inputs are the u∗ uncertainty calculated by Pastorello et al. (2020), and observed daily NEE. Outputs are the outlier score for

each data point and the estimated median upper limit of tolerated u* uncertainty used for flagging site-variables (see section
2.4.7). The difference of the 84[th] and 16[th] percentiles (cf 1 standard deviation for a normal distribution) of the estimated NEE
distribution by Pastorello et al. was chosen as u* uncertainty metric U. The calculation of the outlier score follows as described
in section 2.4.1 but with the modification that heteroscedasticity is modelled as a function of observed (instead of predicted)
NEE and that only positive deviations are penalized since U is strictly positive (in contrast to residuals):

$$O_i^{u* = \frac{U_i - P75_i}{2(P75_i - P50_i)}} \tag{5}$$

The median of $P75_i$ is used for flagging site-variables as it can indicate sites where the u* uncertainty is unusual large.

### 2.4.5 Calculation of the inconsistency score

We sort the vector of outlier scores assigned to a target variable for each data point in descending order and normalize them
by our consistency strictness parameter $n_{IQR}$ such that outliers would be indicated with outlier scores > 1. The result we denote

as $O_i$* where the first value, denoted [1] is from the constraint with the largest outlier score, the second value corresponds to
the second largest outlier score, and so on. The inconsistency score (I) is calculated when we have outlier scores from at least
2 constraints:

$$I_i = \begin{cases} O_i^*[2] \ if \ length \ of \ vector = 2 \\ max(O_i^*[2], 2O_i^*[3]) \ if \ length \ of \ vector > 2 \end{cases} \tag{6}$$

Taking the maximum of the second largest, and twice the third largest outlier score was chosen because occasionally three

410 constraints show consistently elevated outlier scores while the second largest does not exceed the niqr threshold. For example,
considering the choice of niqr=3, an inconsistent data point is flagged if two constraints show residuals outside the fence for
niqr =3 or if 3 constraints show residuals outside the fence for niqr=1.5. Doubling the third largest outlier score is a heuristic
choice – an objective scaling might be derived theoretically when making assumptions about the distribution of (normalized)
residuals and independence among constraints. The calculation of the inconsistency score deals with heterogeneity in data

415 availability (missing data) and associated gaps in the outlier scores because it can be readily computed for when we have two,
three, or four outlier scores available per sample, i.e. not all constraints need to be available all the time.

### 2.4.6 Attribution for flagging hard constraints

This step is relevant only if a) an outlier score from a hard constraint exceeds niqr (e.g. an outlier in the SW_IN vs PPFD_IN
constraint), and if b) none of the target variables assigned to this constraint (here SW_IN and PPFD_IN) were flagged for this

420 data point. This can happen when all the inconsistency scores for this data point and the target variables are below 1 (e.g. I[SW_IN]
< 1 and I[PPFD_IN] < 1), e.g. when all other constraints assigned to target variables give outlier scores < niqr or when no other




constraint was available for the data point. The simplest solution would be to flag both variables, while we want to avoid this to retain as much data as possible.

We use an iterative attribution scheme that aims at identifying and flagging which of the target variable is more likely to show an issue. In each iteration, we loop over the hard constraints and identify and handle only those data points that are outlier points in the current hard constraint but where none of the assigned target variables were flagged yet. Each iteration also executes steps (1) and (2) described in section 2.2.4, i.e. the propagation of flags to dependent variables and the consecutiveness constraint. Since each iteration causes flagging, the number of non-attributed outliers of hard constraints decreases with iteration.

In the first iteration, we force flagging for hard constraints that were assigned only to one variable. For example, outliers of the relationship between $NEE_{NIGHT}$ and $RECO_{NIGHT\_DT}$ are attributed to RECO_DT only. The flag will be propagated to GPP_DT due to dependencies. Applying the constraint on consecutiveness will reject further data points in RECO_DT and GPP_DT. In the second iteration, we inspect if the inconsistency scores for the assigned target variables deviate by more than 0.5. If so, the target variable with largest inconsistency score is flagged. For the example on the SW_IN vs PPFD_IN constraint in Figure 3, the second sample shows $I^{SW\_IN} = 0.8$ and $I^{PPFD\_IN}=0$ causing flagging for SW_IN. In the third iteration, we flag those variables where the inconsistency score exceeds 0.5 – this corresponds to relaxing the specified niqr threshold to half (e.g. to 1.5 from 3). This step catches conditions where e.g. $I^{SW\_IN} = 0.7$ and $I^{PPFD\_IN}=0.3$ where no attribution was done in the previous iteration because the difference of inconsistency scores was smaller than 0.5. In the last iteration, we flag the remaining outlier data points for all variables assigned to the hard constraint.

**2.4.7 Outlier scores for site-variables**

The calculation of the outlier score for a constraint that is attributed to entire site-variables follows the principle of the box-plot rule modified for accounting of asymmetric distributions. Here we also need to distinguish constraints according to whether outliers are in the upper or lower tail. For example, a data issue is indicated by an unusual low correlation (and not by a high one) and by an unusual high fraction of rejected values (and not by a low one).

$$O_s^{lower} = \frac{P25-v_s}{2(P50-P25)} \tag{7}$$

$$O_s^{upper} = \frac{v_s-P75}{2(P75-P50)} \tag{8}$$

Where P25 and P50 are the 25th and 50th percentile of the distribution of values v and s is the index for site. Except for the correlations whose outlier score is only sensitive to the lower tail, all other metrics used for flagging site-variables are only sensitive to the upper tail.

The performance of a machine learning or bivariate constraint is measured by correlation, in a robust way since outliers were removed before calculation (see section 2.4). Occasionally, low correlations are due to very low variance, e.g. in the absence of a seasonal cycle, which we account for by increasing the correlation value passed to the outlier score calculation under conditions of low variance (see below). This conditional upward adjustment of the correlation value is achieved by truncating





at a specified minimum variance (VAR*), which is calculated from the median r2 and the median MSE observed across sites
for low variance conditions (Eq. 10). We chose the 7$^{th}$ percentile of variances to identify low variance conditions. Please note
that this adjustment of the correlation is only relevant for a very small percentage of sites, and that its effect is a decrease of
the outlier score compared to no adjustment.

$$r_s^* = \sqrt{1 - \frac{MSE_s}{max\left(VAR_s, VAR^*\right)}} \quad\quad\quad (9)$$

$$VAR^* = \frac{MSE^*}{1 - median\left(r_s^2\right)} \quad\quad\quad (10)$$

Where $MSE^*$ is the median MSE where $VAR_s < P7\left(VAR_s\right)$.

The outlier score for the regression lines of the bivariate constraints is based on slope and intercept of the linear orthogonal
regression line calculated after removing outliers for robustness (see section 2.4.3). Orthogonal regression minimizes errors
perpendicular to the regression line, i.e. in both X and Y direction, such that slope and intercept are not sensitive to the choice
for X or Y. The outlier score based on slope and intercept is based on first calculating the mean distance of a site to all other
sites, and then the outlier scores sensitive to the upper tail (Eq. 8) is calculated using the distribution of distances (Figure 6).
Because the values of slope and intercept are not comparable and have different units, the Euclidean distances are calculated
in two dimensional space where the two dimensions is a pair of values that define the regression line: 1) Y at x=0 which is the
intercept, and 2) Y at a typical value of $x=x_a$, i.e. for the pair ($b_s$, $x_a*m_s+b_s$). $x_a$ is calculated based on the (robust) range of
intercepts and the slopes for a given constraint:

$$x_a = median\left(\frac{P99(b_s) - P1(b_s)}{m_s}\right) \quad\quad\quad (11)$$

The final between-site outlier score for a bivariate constraint is then the maximum of the regression line based outlier score
and the correlation based outlier score. The maximum corresponds to a logical OR operation, i.e. outliers appear if the bivariate
constraint shows unusually weak performance or its regression line is unusual. Using both outlier scores individually as a
constraint instead of their maximum composite would cause problems due to violating the requirement of independence among
constraints.

### 2.4.8 Detection of temporal discontinuities

Temporal discontinuities are assessed per target variable and site. Additionally to removing gaps (fqcOK < 0.8), we also
remove flagged data points for the target variable (see section 2.2.4). Then, we modelled the entire time series of the target
variable with two Random Forests: (1) using SW_IN, PPFD_IN, SW_IN_POT, TA, VPD, and water balance indicators (see
section 2.4.2) as predictors to account for variations in weather (the target variable was removed from the predictor set when
relevant), and (2) using only SW_IN_POT and day of year as predictors to account for seasonality only. We use the cross-
validated predictions of these models to calculate the residuals (cf section 2.4.2), and normalize those by the estimated IQR of
the residuals varying with magnitude of the predictions respectively (Fig. 7, see also Figure 1 and 8). The distance based test
statistic from (Szekely and Rizzo, 2004) for the difference of two distributions X and Y (here from two temporal segments) is:





$$T = \frac{n_1 n_2}{n_1 + n_2}\left(\frac{2}{n_1 n_2}\sum_{i=1}^{n_1}\sum_{m=1}^{n_2}||X_i - Y_m|| - \frac{1}{n_1^2}\sum_{i=1}^{n_1}\sum_{j=1}^{n_1}\left||X_i - X_j|\right| - \frac{1}{n_2^2}\sum_{l=1}^{n_2}\sum_{m=1}^{n_2}||Y_l - Y_m||\right) \tag{12}$$

Where n1 and n2 denote sample sizes for X and Y respectively, and || denotes Euclidean distance. The term in the brackets equals the energy distance between the distributions of X and Y, which measures two times the mean distance among samples between X and Y minus the mean distance among samples within X and within Y.

We did not calculate the significance for the test statistic, i.e. whether the difference in distributions is significant, because we found that it is not useful for our purpose: the significance test almost always returned significance while being computationally very expensive due to the necessity of recalculating the test statistic for many random permutations. To obtain a measure for the severity of a temporal discontinuity in a time series that is comparable across sites and variables we calculate the change in relative segment dispersion associated with each split. Dispersion is the sum of distances among data points (i.e. the sum of the distance matrix). Relative segment dispersion D is the sum of within segment dispersion $D_l$ over all segments  normalized by the initial dispersion $D_0$ among all data points (i.e. before segmentation).

$$D = \frac{\sum_{l=1}^{n_L} D_l}{D_0} \text{ with } D_l = \sum_{i=1}^{n_l}\sum_{j=1}^{n_l}\left||X_i - X_j|\right| \tag{13}$$

At the beginning and before the first split, D=1. With every split z, D decreases and our break severity measure is the difference in D associated to the split compared to before the split ($\Delta D_z = D_{z-1} - D_z$). For example, consider that with the first split D decreased from 1 to 0.7, i.e. by 0.3, while for the second split it decreased from 0.7 to 0.68, i.e. only by 0.02 – the first split was obviously much more severe and relevant compared to the second.

After the break point detection was run for all sites and all target variables we estimate the outlier score (sensitive to the upper tail, Eq. 8) for every split based on the distribution of ΔD pooled for all sites and variables. This helps judging on how unusual a detected break point is given the context of the site network.

## 3 Results

### 3.1 Patterns of flagged data

Running the C2F algorithm across all sites in FLUXNET 2015 we find most flagging for GPP and TER, followed by SW_IN, NETRAD, and LE, and comparatively few rejections for H, NEE, TA, and VPD (Figure 9). These differences in the fraction of flagged values do not entirely reflect a gradient of data consistency but can also be influenced by the number and quality of constraints available for the different variables (see section 4.1.1). Increasing the consistency strictness from more loose (niqr=3) to more strict (niqr=1.5) can cause more than a doubling of flagging. For GPP and TER the fraction of flagged data exceeds 20% for strict (nIQR=1.5) consistency and is below 10% for loose (nIQR=3) consistency across the full data set, while there is a tendency of slightly more frequent flagging for the daytime based estimates compared to the night-time based. There is substantial variability in the fraction of flagged data between sites (Figure 9 top panel).

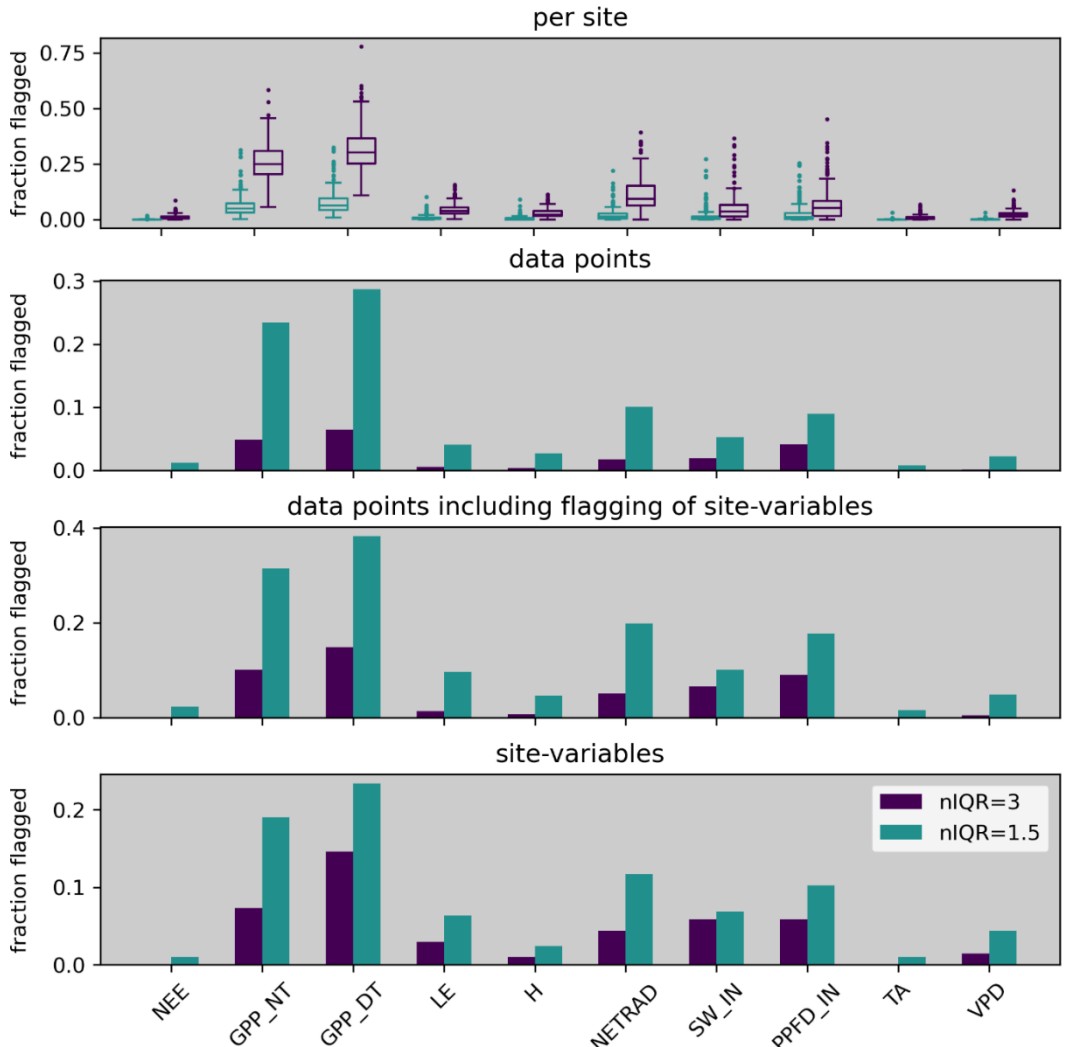

**Figure 9: Summary of the fraction of flagged data for FLUXNET 2015 for loose (niqr=3) and strict (niqr=1.5) consistency. The top panel shows the distribution across sites. Second panel shows the fraction of flagged data points across the full FLUXNET 2015 data set while not considering when an entire site-variable was flagged – these data points are included in the third panel. The bottom panel shows the fraction of sites for which an entire variable was flagged.**

We take a closer look at GPP and TER flagging in order to better understand the pattern of frequent flagging. Figure 10 shows a systematic pattern of flagged GPP and TER values when temperatures are high and GPP is low. These conditions correspond typically to very dry conditions, where the assumptions of the NEE flux partitioning methods are more frequently violated: ecosystem respiration is less controlled by temperature, and GPP is less limited by light. Visual inspection of the time series (e.g. Fig. 5) suggested particular flux partioning issues during respiration rain pulses, where e.g. GPP_NT is often systematically negative – a pattern of strongly elevated flagging frequency during and after rain was observed when temperatures are high (>15°C) and GPP is low (Figure 10). This illustrates methodological limitations of the flux partioning





methods in dealing with rapid changes of ecosystem responses due to the used moving window approach to estimate parameters. To verify that the systematic patterns found for flagged GPP and TER values under high temperatures and low GPP conditions is not an artefact of the method we compare them with the patterns for LE and H, where we essentially see no systematic patterns in the relative frequency of flagged values along with an order of magnitude smaller fraction flagged.

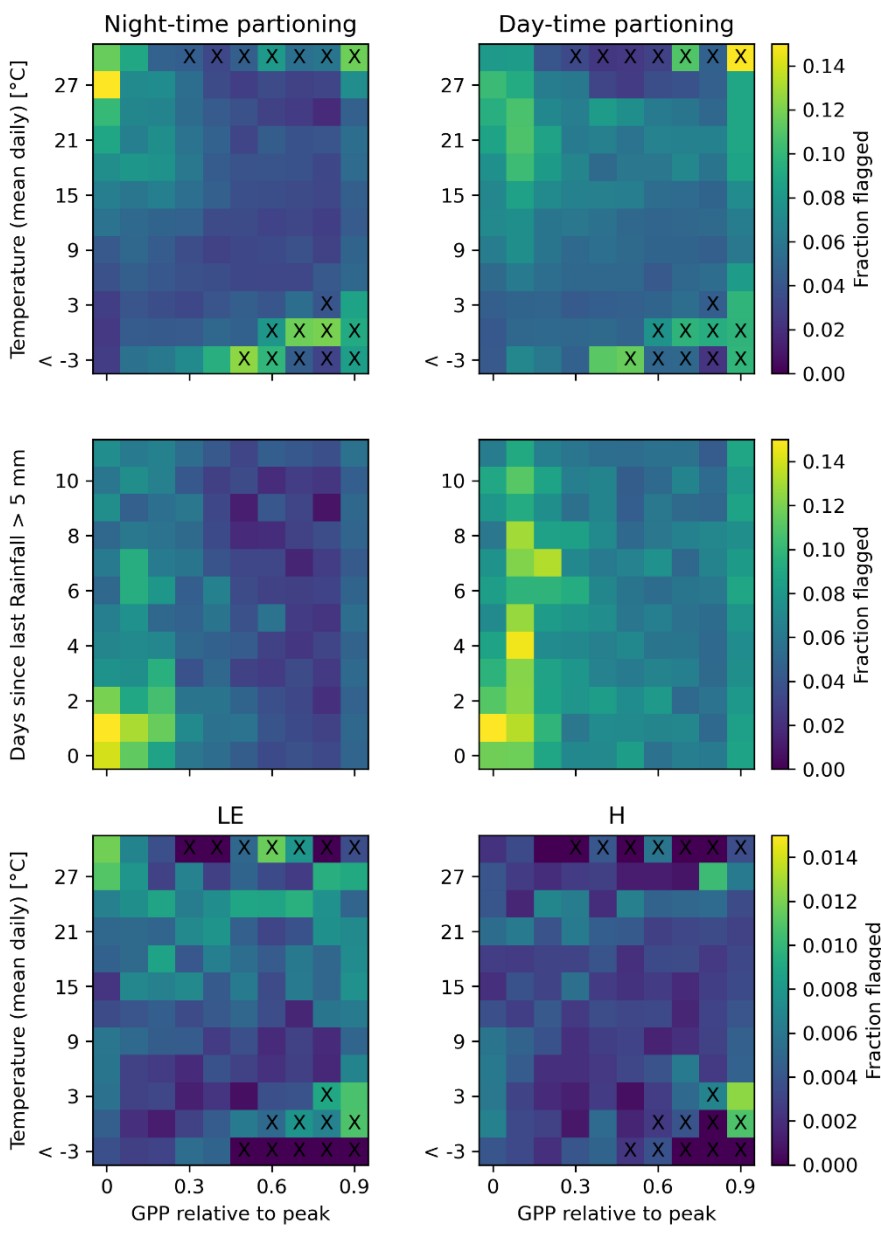


**Figure 10: Relative flagging frequency for nighttime and daytime flux partitioning (GPP, Reco, top row), latent energy (LE) and sensible heat (H, bottom row) as a function of daily temperature and relative GPP. The middle row shows flagging patterns for GPP and Reco as a function of days since last major rainfall event, where only data with daily temperatures > 15°C were included. Crosses indicate very rare occasions (< 0.05% of data points in the bin).**





We assess whether the flagging of the entire GPP or TER variables for sites also follows a systematic pattern, and find that

there is indeed some prevalence for flagging for sites with low mean annual precipitation and at high mean annual temperature

(Figure 11). A similar pattern is not clearly evident for other variables, except for the flagging for NETRAD at very cold sites.

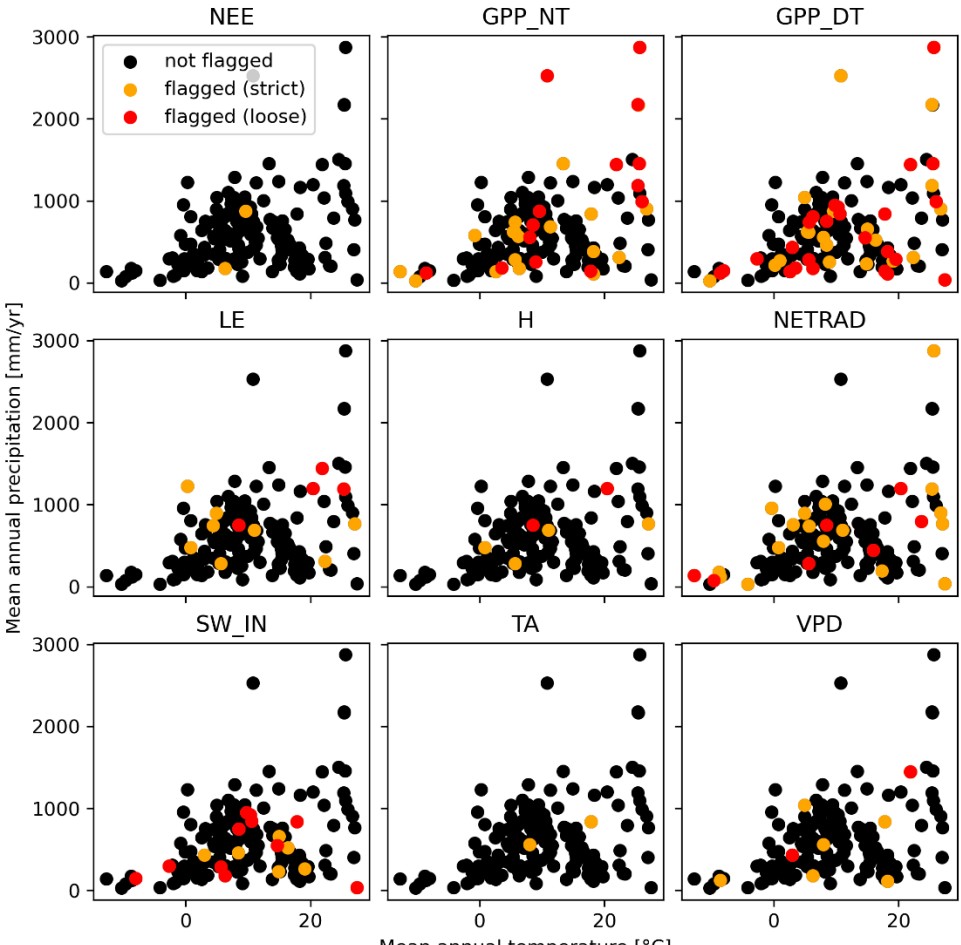

**Figure 11: Flagged site variables in mean temperature and precipitation space. Each dot corresponds to a site.**

We now assess to what extent flagging might be systematic for "extreme" conditions recorded in the time-series of the sites.

We chose to look at cold, normal, and hot conditions because temperature extremes are a common topic of interest and because

the temperature variable showed least data consistency issues. The boundaries for the temperature extremes were chosen

according to the boxplot rule for the distribution of measured daily temperatures at each site with a threshold of 1.5 units of

interquartile range in terms of distance from the median. Overall, we see no evidence that the C2F would be flagging a high

fraction of data points related to extreme temperatures (see y axis values in Fig. 12). However, for some variables we see a

larger fraction of flagged values for extreme temperatures compared to normal. Relatively more frequent flagging for GPP

under hot conditions likely reflects primarily real data issues related to violations of flux-partitioning under drought as outlined

above. There is a similar pattern for LE, while it is important to note that the fraction of flagged values under hot conditions is





still very small (less than 2%). SW_IN and NETRAD also show elevated flagging rates at high temperatures while the vast
majority of data is still retained. For H and TA, flagging rates are increased at cold and hot conditions compared to normal but
are small on an absolute level. Overall, we conclude that there are some indications for more frequent flagging under extreme
conditions. At the same time the percentages still remain to be small suggesting that the C2F procedure is generally robust to,
and not very biased at extreme conditions. The slight tendency of elevated flagging percentages at extreme temperatures might
be related to limitations of estimating heteroscedasticity in very data sparse conditions (see section 4.1.1).

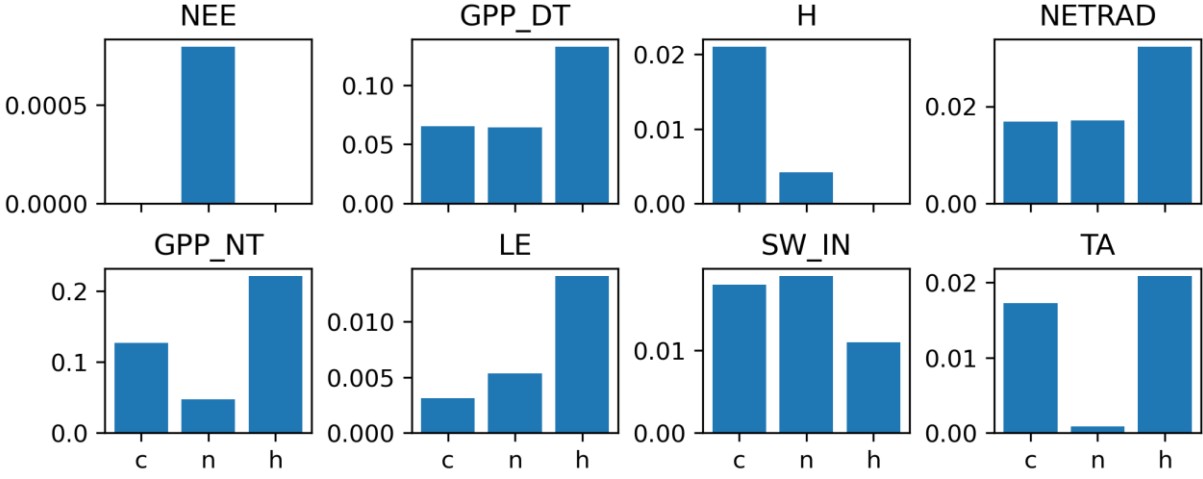


**Figure 12: Relative rejection frequency for cold, normal, and hot temperatures corresponding to the loose consistency criterion (niqr=3). The stratification in temperature classes is based on the boxplot rule applied separately for each site with niqr=1.5.**





### 3.2 Patterns of large temporal discontinuities

We illustrate in Figure 13 for some sites that large discontinuities in the time series are detected for ecosystem fluxes and meteorological variables, which often coincide with documented changes in instrumentation (BADMs) or changes in the ecosystem. Changes in instrumentation can explain detected discontinuities e.g. at IT-SRo in 2007-2008 for NEE and LE, at CH-Dav in 2005 for NEE and LE, at BE-Lon in 2007 and 2012 for NEE, at CA-TP1 in 2008 for NEE, at NL-Loo in 2004 for NETRAD, at AU-Tum in 2008 for NETRAD, and for IT-Col in 2005 for SW_IN. Changes in the ecosystem have likely caused

detected discontinuities at DE-Tha for NEE in 2002 (thinning) and at FR-Pue for NEE in 2005 (thinning). No discontinuity was detected for the long NEE time series at DE-Hai indicating that the method is quite robust to (real) interannual variability caused by weather. The reason for the many discontinuities of NEE at BE-Lon is likely that it is a site with crop rotation, while some detected discontinuities coincide with changes in instrumentation. Likewise, CA-TP1 is a growing forest plantation established in 2002 with an associated strong trend in ecosystem structure which could explain the detection in 2008, while

this coincides also with changes in instrumentation. Time series patterns suggest that detected discontinuities at FI-Sod for H and at IT-BCi for TA are also likely due to changes in instrumentation, while those were not reported in the BADMs.

There are several instances where changes in instrumentation are not associated with the detection of a temporal discontinuity (e.g. FI-Sod in 2003), and there are likewise several detected discontinuities, which we can not associate with documented changes in instrumentation or ecosystem type. These considerations highlight the importance of correct and complete meta-

data on instrumentation and ecosystem changes for interpreting time series of flux tower measurements and detected discontinuities.





**Figure 13: Illustration of detected breaks for some sites and variables. Black vertical bars correspond to breaks exceeding niqr=1.5 (strict); an additional change in color corresponds to bigger breaks (niqr=3, loose). Dates of changes in instrumentation recorded in BADMs are labelled as dotted cyan lines (sonic anometer), magenta triangles (gas analyser), and white triangle (measurement height). Other changes where only the year was given is shown as text.**





Across FLUXNET 2015, large discontinuities (niqr>3) in LE are detected for about 25% of sites, in NEE, SW_IN and
       NETRAD for about 20% of sites, and in H and TA for about 10% of sites (Fig. 14). Considering very big discontinuities (up
       to niqr=6 and larger) we see that the fraction of affected sites levels off at about 15% for LE, SW_IN and NETRAD suggesting
       that also changes in the instrumentations of radiometers may be causing more frequent discontinuities in the data than perhaps
       anticipated. We find that for the majority of long-term sites, at least one big discontinuity was detected for the radiation fluxes,
LE and NEE.

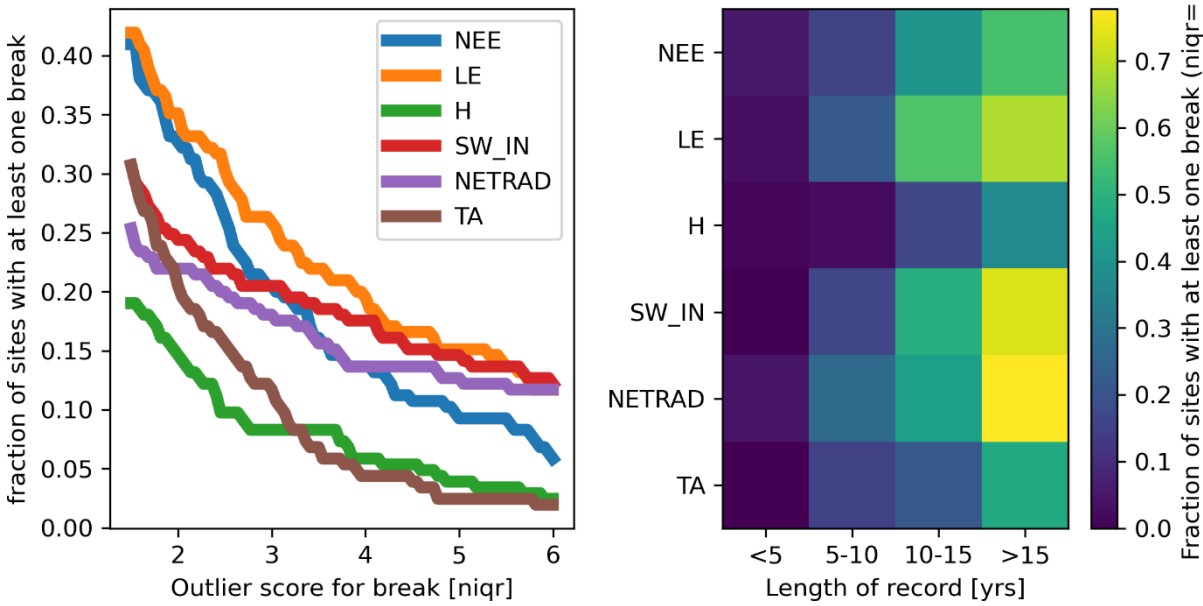

**Figure 14: Fraction of sites with at least one break detected for different variables as a function of the applied outlier score threshold (left), and as a function of record length (right) based on niqr=3.**

## 4 Discussion

### 4.1 Methodological considerations

The key objective of any data screening approach is to distinguish between appropriate and inappropriate data, while there is
       some arbitrariness and context dependence in the definition of what is appropriate. We addressed this aspect from a conceptual
       point of view by flagging data that are inconsistent based on multiple expected relationships among variables, where the
       strictness on the inconsistency definition can be varied by the user according to specific demands and applications. The key
question is on the effectiveness on the C2F algorithm in flagging ideally all inappropriate data while retaining ideally all
       appropriate data. The key challenges in assessing this are the lack of reference or validation data for inappropriate data, and
       that inappropriate data are expected to be rare cases. In the following, we discuss methodological aspects related to erroneously
       flagging appropriate data (false positive) and erroneously not flagging inappropriate data (false negative).



### 4.1.1 Factors for potential false positive and false negative flagging

C2F is fundamentally based on the distribution of residuals from expected relationships among variables rather than on the distribution of the target variable itself. The latter is a common quality control procedure of identifying outliers, e.g. based on the box-plot rule, but flags extreme values in the tails of the distribution by construction risking false positive and systematic flagging. Thus, our choice of working with residuals is preferable here because values in the tails of the target variable distribution are retained as long as deviations from expected relationships are not extreme. We showed that C2F is effective

in retaining data under extreme temperature conditions (see section 3.1).

Hence, it is relevant to assess whether detected inconsistencies, i.e. large deviations from expected relationships, can also be real rather than pointing to data issues. We addressed this aspect by distinguishing between hard and soft constraints (see section 2.2.1), where for soft constraints large deviations do not force flagging of values immediately. Most soft constraints are related to ecosystem fluxes (see Table 3) because e.g. functional changes in the ecosystem may change relationships

between meteorological variables with the fluxes as well as relationships between different fluxes. Flagging is only enforced when more than one soft constraint indicates an inconsistency, while it is important to construct the different constraints to be independent from each other as much as possible. This approach essentially tries to minimize false positives and tries to avoid flagging data that appear as unusual but might be real, while this obviously comes with the risk of having more false negatives. The consideration of heteroscedasticity of residuals from a constraint has been key in minimizing false positives and negatives.

The fact that the variability of residuals typically increases with magnitude (Richardson et al., 2008) implies that we would get many false negatives at low magnitude and many false positives at high magnitude leading to a severely biased and systematic pattern of flagging if not accounting for heteroscedasticity (see Fig. 8). In the estimation procedure of the heteroscedasticity it was important to extrapolate the distribution properties of the residuals to the tails of the distribution of the target variable where they cannot be estimated empirically in order to minimize false positives there. This was not accounted for in previous

methods based on binning residuals based on magnitude (Vitale et al., 2020). Indications of elevated flagging frequencies under extreme temperature conditions (Fig. 12) may indicate some remaining uncertainties in accounting for heteroscedasticity, which is particularly challenging at the tails of distributions due to data scarcity.

The estimation of the outlier score is an adaptation of the boxplot rule, accounting for potential asymmetries in the distributions. This is preferable over excluding a fixed percentage of data that is sometimes done. By varying the nIQR parameter we can

choose how strictly we apply C2F as this determines how far into the tails of the distribution of residuals a data point is allowed to fall. Increasing nIQR makes C2F becoming more loose leading to less flagged data overall, less false positives but more false negatives. Choosing the boxplot rule is common practice of identifying outliers as it is simple and avoids making assumptions about the underlying distribution. However, the expected probability of a data point in the tails of the distribution is certainly dependent on the specific properties of the distribution, in particular of the skewness and kurtosis, which are very

hard to estimate empirically in a robust way (Ritter, 2023). In addition we did not account for sample size corrections of the





boxplot rule (Ritter, 2023; Schwertman et al., 2004) in the current version since these are also sensitive to the assumptions of the underlying distribution. All these factors cause some uncertainty for false positives and false negatives.

Being able to define meaningful constraints is a prerequisite for C2F to work. For target variables where we have less constraints, like here e.g. for TA and NEE, more false negatives must be expected. Practical issues due to the lack of data in evaluating a constraint obviously increase the risk of false negatives. Likewise, issues due to non-stationary behavior of relationships due to e.g. changes in instrumentation increase the risk of false negatives because the overall distribution of residuals will be wider and thus more forgiving. Running C2F within segments identified by the detection of temporal discontinuities could improve this aspect in the future. Overall, we can expect that the more appropriate the flux tower data are already before we apply C2F the better and more precise C2F works in identifying remaining inconsistencies.

### 4.1.2 Detection and interpretation of discontinuities in the time series

Our detection of temporal discontinuities is based on model residuals rather than on the raw data of the flux tower variable. This is preferable because (1) the data typically show very large seasonal variations that would propagate to the test statistic, and (2) gap-filling of long gaps is not needed, which is particularly challenging in this context because it would require filling with realistic variability (including noise) to avoid artefacts in the distribution of the data.

We chose to use residuals from two models jointly in the estimation of the breaks: (1) a machine learning model based on environmental conditions and (2) a machine learning model based only on the day-of-the-year and potential shortwave radiation that effectively performs a deseasonalization. The advantage of the first model is that it accounts for observed variations due to changes in environmental conditions. However, we found that the machine learning model was too flexible in some instances where the model predicted an obvious artefact in the distribution of a target variable because there was a concomitant change in one of the predictors. The latter can happen because the predictor variables are also flux tower data, e.g. if two instruments are modified at the same time, or if the tower is moved or raised. This was the reason to additionally include the deviations from seasonality obtained from the second model. The residuals of both models were normalized to account for heteroscedasticity, which further minimizes differences in distributions due to different proportions of different seasons when calculating the test statistic.

Currently, discontinuities are flagged based on how unusual differences in distributions are using an outlier score calculated from a distribution of break severities pooled across all variables. While our definition for flagging temporal discontinuities is simple and allows for varying the threshold, it is clear that it is not directly related to whether a detected discontinuity is meaningful or relevant for a certain application. Further, pooling the distribution of break severity across all variables rather than evaluating per variables has the advantage that it allows for (1) obtaining a larger sample size to better characterize the distribution, in particular its tail, and (2) better comparability among variables in terms of which variables are more affected by breaks. At the same time, pooling across variables is not ideal from the perspective of hunting artefacts due to instrumentation changes since we expect more false positives for ecosystem fluxes compared to meteorological variables due to the possibility of real disruptions in the ecosystem.



The breakpoint detection is based on assessing changes in the distribution of residuals from machine learning models. That
implies that any factors impacting the target variable distribution that were not accounted for in the modelling can elevate the
test statistic. Beyond abrupt changes in instrumentation that we would like to flag ideally, natural reasons for a change in the
distribution of residuals can be natural or anthropogenic disturbance like events (e.g. insect outbreaks, fires, windthrows,
harvest, thinning, crop rotation, other management practices) that change ecosystem properties. Also gradual changes in the
distribution of residuals could in theory cause the detection of a break, e.g. due to strong trends in (1) ecosystem properties,
e.g. due to succession or post-disturbance recovery, (2) environmental conditions that are not modelled (e.g. $CO_2$ fertilization),
(3) target variables due to drifting sensors.

From our results applied to FLUXNET 2015 we have some indications that the effect of trends does not cause a large proportion
of flagging discontinuities. While we expect a trend in air temperatures in many long-time series due to global warming, we
find that the air temperature variable is among the variables that is least affected by detected discontinuities (see section 3.2),
probably because air temperature is comparatively easy to measure.

Our results further suggest that detected discontinuities due to instrumentation artefacts seem to dominate over natural, real,
changes. We see for example relatively large differences in frequencies of big discontinuities between radiation fluxes and
temperature (Fig. 14), or between LE and H. Such large differences would be unexpected if detected discontinuities were due
to real environmental changes. Instead these differences in the frequency of detected breaks among variables corresponds to
different levels of complexity for measuring variables: While temperature sensors are robust, long lasting, and require little
maintenance, pyranometers are sensitive to levelling, position, and contamination requiring more frequent maintenance and
replacement. Sensible heat is measured by the sonic anemometer directly which typically runs for years without major
problems, while for latent energy the infrared gas analyser requires more often calibration, maintenance and replacement – the
same applies for the tube of closed path systems.

Clearly, some of the detected discontinuities are due to real changes as illustrated for some examples (Fig. 13). This implies
that detected discontinuities require careful attention in order to judge whether it is due to an artefact or a real phenomenon
and it confirms the importance of complete metadata and ancillary data as crucial set of information for the proper interpretation
of the measurements.

### 4.2 Notes and recommendations for applications

### 4.2.1 Flagging inconsistent data

Adding or modifying constraints or the strictness parameter nIQR for custom applications is straightforward. Since
almost all computational costs are associated with calculating intermediate diagnostics that are stored it is straightforward and
fast to obtain new flagging results for modified consistency strictness. This is particularly useful for assessing the relevance of
the data quality-quantity trade-off for the conclusions of a specific application. If that is desired, we recommend to recalculate
the flags for nIQR varying from 1.5 to 5, e.g. at intervals of 0.1, and finding the smallest nIQR at which flagging is indicated.



This yields a more continuous representation of inconsistency and facilitates straightforward filtering. The recalculation of flags for different consistency strictness is preferable over using the inconsistency score because the flagging takes additional considerations into account (see section 2.2.4).

The approach for daily data outlined here can in principle be adapted to sub-daily data but it would require modifying some of the constraints and it may become computationally very expensive. The latter was the reason to stick to daily data here. For applications using sub-daily data we recommend to discard all sub-daily data of a flagged daily value for now.

C2F delivers flags for individual data points for a variable and site as well as flags for entire site-variables. While it is not feasible to scrutinize every flagged data point to make a decision if one wants to include this or not, we suggest that scrutinizing the flagged entire site-variables manually is feasible and recommended, i.e. the flagging of site-variables is only meant to draw
the attention to potential data issues that require further investigation. This is particularly relevant because e.g. GPP from sites at the fringes of the tower distribution in climate space are more frequently flagged and these sites are in principle particularly precious for global synthesis studies.

### 4.2.1 Flagging temporal discontinuities

The detected discontinuities in flux tower variables are meant to draw the attention to potential artefacts in the data, which
then requires further investigation and judgement depending on the application and data needs. In particular, discontinuities in ecosystem fluxes can be due to real changes of the ecosystem, e.g. due to disturbance, harvest, crop rotations, or other management practices. While it is hard to formalize, we can provide some guidance on collecting indications which of the reasons may apply from logical reasoning. For a disturbance like event in the ecosystem we expect breaks in several ecosystem fluxes around the same time but not for meteorological variables like SW_IN and TA, or at least much less severe changes. In
addition, we expect that a disturbance would shift the ecosystem NEE towards less carbon uptake on average. Detected discontinuities around the same time in ecosystem fluxes and meteorological variables may indicate a major change in the instrumentation infrastructure such as raising or moving of the tower, or changing the infrared gas analyzer. To infer whether a flagged discontinuity is due to a trend in the variable one could simply remove the trend in the residuals before inputting them to the calculation of the test-statistic.

How to deal with detected discontinuities in the time series can also be very application dependent and may vary from discarding the site, keeping only the longest segment, running the analysis separately within segments, or not doing anything about it. Clearly, analysis targeting interannual variations or trends should consider discontinuities in the time series that could be artefacts of changes in the measurement setup.

### 4.3 Flagging patterns

Applying C2F to FLUXNET 2015 has revealed three major patterns of data inconsistencies: (1) comparatively large flagging frequencies for GPP and Reco with a systematic pattern of more frequent flagging under dry - hot conditions, especially after





rain; (2) comparatively frequent flagging for SW_IN and NETRAD; and (3) frequently detected discontinuities in long time series of LE, NEE, SW_IN and NETRAD.

### 4.3.1 Flagged data points

The high proportions of flagged GPP and Reco data under drought conditions can be because the relationship between night-time respiration and temperatures break down, or because fast rain pulse responses get obscured by the moving window approach used for NEE partitioning. For the daytime partitioning we see a tendency of more flagging at high temperatures and higher GPP compared to the night-time method. Potentially this is due to imperfect accounting of the VPD effect on GPP in the parametrized light-response curves. While the absolute values of GPP and TER are often quite small under such dry
conditions this issue causes comparatively little uncertainty for annual budgets. However, they imply some limits to our ability to better understand ecohydrological functioning under water stress and rain pulses. Novel flux partitioning methods like (Tramontana et al., 2020) that take water stress conditions better into account and avoid fitting in moving windows would be important complements to have in the near future.

Relatively frequent inconsistencies in radiation variables may be due to issues in correctly installing, calibrating, and
maintaining the pyranomaters. Radiation data are crucial for the interpretation of ecosystem fluxes, and are required as forcing variables for models. Furthermore, NETRAD is often used to estimate evaporative fraction as a water stress indicator, and needed for analyzing or correcting the energy balance closure gap problem. This calls again to the importance of the maintenance of the sensors and the correct and full recording and reporting of all sensors replacements or calibrations in the metadata.

### 4.3.2 Flagged discontinuities in time series


We found that most long-term sites show discontinuities for radiation variables, LE, and NEE. Obviously, this might have large implications for studying many outstanding questions regarding interannual variations and trends of ecosystem fluxes using FLUXNET. While such discontinuities can also be due to real changes in the ecosystem we have indications that data issues are likely the prevalent reason (see section 3.2). Even if half of those would be attributable to 'false alarm' in a very
conservative scenario, it would still represent a very relevant problem for the community.

Comparatively rarely detected discontinuities in TA and H could be because the associated instruments are quite robust and long-lasting. In contrast, pyranometers need replacement and maintenance more frequently, are subjects to drifts, which could explain more frequently detected discontinuities in radiation variables. For LE and NEE in particular, changes in the gas analyser, and for closed path systems changes or maintenance of the tube, or a change of the spectral correction used by the
principal investigator (PI) could cause temporal discontinuities. It would be important to better understand what aspects related to instrumentation and maintenance change are causing the main problems here to facilitate consistent long time series in the future. Also, in this case the availability of metadata about sensors or setup change or major disturbances/management activities



at the sites are very important for the interpretation of detected discontinuities and could allow for more tailored approaches in the future.

**5 Conclusions**

Using expert knowledge and experience we designed and implemented C2F, a complementary data screening algorithm for flux tower data based on the principle of detecting inconsistencies. It is fully automated, transparent, follows objective principles, and delivers simple Boolean flags that are straightforward to use.

Clearly, C2F is not perfect, complements and cannot replace the typical quality control of flux tower data done by PIs and
during standardized processing. In fact, it relies on the assumption that the vast majority of data are appropriate already. The quality of our flags also relies on data availability in terms of variables, i.e. the number of constraints that can be used, and data quantity for robust estimation of the statistical metrics used. To further develop and improve C2F it would be desirable to be able to benchmark it objectively using a large set of synthetic data, where flux tower data with all its potential issues and noise properties are realistically emulated with labels for inappropriate data available.

Applying C2F to the FLUXNET2015 dataset uncovered for instance issues of the NEE flux partitioning into GPP and Reco under dry and hot conditions, as well as temporal discontinuities in long-time series of e.g. LE and NEE. While the potential existence of such problems is no surprise for eddy covariance specialists, C2F provides associated flags, which were not available before. This is especially useful for synthesis activities, ecosystem modellers, or remote sensing integration with machine learning. We therefore hope that C2F helps in making scientific progress, helps in improving FLUXCOM and
process-based models, and helps in flux tower data becoming more accessible and used across communities. In addition, it could help in assisting PIs to assess data consistency before submission to regional networks, and it could help in accelerating feedback loops between PIs and centralized processing units of regional networks, if C2F would be implemented in ONEFLUX.

**Author contributions**

MJ developed and implemented the methodology, performed the analysis and drafted most of the manuscript. JN helped with coding and applied the code to FLUXNET data. JN and TW are developing code for the a user interface. MM, TEM, DP, TW and MR contributed expert knowledge on eddy covariance quality control. All authors provided intellectual input to the work and manuscript.



**Code availability**

The code is available on request during discussion phase. After revision and potential modifications it will be available on github along with a simple user interface.

**Data availability**

The flagging results are available on request during the discussion phase. After revision and potential modifications, the data be publicly available.

**Competing interests**

The contact author has declared that none of the authors has any competing interests.

**Acknowledgements**

The authors acknowledge funding from EU H2020 projects CoCO2 (GA 958927) and E-SHAPE (GA 820852), and from the European Space Agency for the SEN4GPP project and the Living Planet Fellowship Vad3e mecum.

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
