# Peer review of "Technical Note: Flagging inconsistencies in flux tower data"

_Biogeosciences, 2023_

## Referee Comment (RC1)

Technical Note: Flagging inconsistencies in flux tower data

This is definitely a technical note, and as long as the editors of this journal are willing to consider it, I will give it due diligence as a referee. Otherwise, this document could be used as a white paper or grey literature to supplement processing on the ICOS or Fluxnet web sites.

I appreciate the utility of having a set of well defined and accepted flags for data by the community, as data has certain quality due to time and place. Sadly we are aware that many data users who are not involved in the details and rigors of the measurements, processing and interpretation often ignore these flags. But no harm in producing and providing them.

What is unique and distinct here is the production of a complementary set of consistency flags (C2F) for flux tower data, which rely on multiple indications of inconsistency among variables, along with a methodology to detect discontinuities in time series. I am a fan of multiple constraints, so I am willing to read the case before me.

As I read this work I have some disagreements with conditions which they may flag.

For example it is stated that most frequent flags were associated with photosynthesis and respiration during rain pulses under dry and hot conditions. I have spent a good part of my career studying such pulses. They are real and sustained following rain events. To remove them is faulty and will cause biases in sums. Yes, I concur during the rain event itself data such be flagged when sensors are wet. But following the rain, huge amounts of respiration can and will occur.

**2 Materials and methods**

As I read this paper back and forth, I wonder about the wisdom of its organization. In the Methods section there are 8 figures or so. This seems more like a Results and Discussion. I also suspect much of the excess material could be in an Appendix or Supplemental material. For a Technical Note, this paper is really long and excessive.

Section 2.2.1. It is important to benchmark and monitor the relation between PAR and Rg. It is our experience that quantum sensors tend to drift over time, if not frequently calibrated. PAR is used to upscale fluxes with remote sensing and if those relationships are built on faulty values of PAR, the derived products in time, space and trends will be in error. Hence, looking at these flags can have important implications. Too often this issue has been overlooked, so it would be important to know the consequences of improving these data.

The relation between Rn and Rg is important to examine, but realize it will change with season as albedo and surface temperature changes. So be careful and do not make your flags by using one annual dataset for the site.

Comparing GPP and Reco with Day vs Night partitioning methods may be interesting, but not sure which is right. We know there is down regulation in dark respiration during the day, it is hard to measure reliable $CO_2$ fluxes at night under stable conditions and with tall vegetation and appreciable storage and or sloping terrain. These can be points of reference and maybe the daily sum is better than hour by hour measurements, as some errors cancel.

I have learned from Dario the value of plotting $CO_2$ flux vs u* and developed a matlab subroutine to do so. The threshold can be uncertain as u* has some autocorrelation with the flux. Of course we don't

want to set high thresholds as they are based on a diminishing number of data points as high u* values are rare compared to low ones.

I see one of the constraints is LE +H vs Rn..What about G or storage in water column?  I think these tests are only instructional.  We know that there are many differences in sampling areas and representative of radiation and fluxes.  It is dangerous to indict one or the other. And with wetlands it is really hard to measure water storage.  We have a data set with nearly closed energy balance and then flooded the system and it all went to hell. Same sensors, same processing, ideal fetch and site. Just water moves heat in and out and it is hard as hell to sample well and well enough.

I must admit I am having a problem coming up with a salient point of this paper and how it will help me do better. I am at the point where an outlier score is proposed. It seems ok, but it is a lot like the college ratings, that depend upon an arbitrary set of metrics and scores.

I often advise use the set of sites that help you ask and answer the questions you are asking, relating to climate, function and structure. Just because these sites and data are in the fluxnet database does not mean we have to use them all. Maybe this should be the point of this paper.

Figure 1.  It is a comparison between machine learning and flux data. Not sure what I am to learn and extract here.  Which is right or wrong?  Machine learning ultimately is a fancy least squared fit to a bunch of transfer and nodes.

Here is a set of data comparing annual carbon fluxes with machine learning methods from my sites. They are almost indistinguishable from the direct flux measurements they are derived from. In this case we know our site and develop the machine learning model with the most appropriate and representative biophysical forcings.  In the figure given in this paper, I have no idea how appropriate the machine learning model may be for this situation, as the answer is based on independent variables they chose to use or omit.

[Figure]

Regarding the comparison of radiometers I know during some seasons our guy wires may shade the quantum sensor for certain angles of the sun. surely those data are not fit and I hope such a method may help detect these biases and errors.

Figure 2 seems to be a nice case study to show the attributes of your ideas. Maybe start with that one first. It is clear and more understandable, as we know PAR and Rg are closely related. So when there are differences it can help us think about why and which is more plausible and better.

Fig 3. Maybe I am just tired, or thick, but I don't follow the logic and rationale of the flag for light used for GPP. It would only give me pause on the accuracy of the machine learning calculations, but not the eddy fluxes.

Fig 5. I am trying to get my head around the issue of the comparison of the daytime vs nighttime methods. Again, I would argue one is better than the other. Personally, I like the idea of multiple constraints and see if the two methods are converging for confidence, more than anything. Not sure what you all are doing, but in early days working with Eva Falge, we estimated respiration during the day by the extrapolation of the CO2 flux vs light response curve. Now one of the limits is basing a regression and extrapolation on only a few points when the response function is linear, and the fact that during the sunrise sunset period steady state conditions don't hold.  It is these reasons why I argue against one being better or worse, but if they both converge at least we may assume the fluxes may be good enough.

The reality is that pulses due to rain or insects passing through the path of the IRGA or sonic are problematic. Or those from electrical noise (a rarity today). We also see problems with CO2 fluxes over open water as there is a covariance with w and RSSI of the sensor that yields fluxes in the wrong direction and that are not physical. Those should be filtered. But I don't hear about that here.

Fig 6 seems to align with my suggestions that some sites may not be the best for some analyses and just toss them. Nothing lost as we oversample in many situations.

Fig 7. Curious as to why there is a systematic jump in LE. Eddy covariance should be immune from just a jump as we are doing mean removal. So even if sensors change and they are properly calibrated we should not expect such a marked difference. This is not like comparing two separate sensors, that can have offsets.

Fig 8. Illustration of the outlier score. This is needed to support the method described here. Has taken a long time to get to this point. Line 350!

Results

Fig9 demonstrates the point of this method. As expected met variable values tend to have few outliers.

Fig 10 provides a needed diagnostic as to when data may be rejected

Fig 11. Would think this would be a function of open vs closed path sensors

Fig 13. The jumps in NEE seem to be with site management. So Know Thy Site. Just don't blindly process long term data. This is why we have phenocams at our tower, to look at the vegetation when things are 'weird'.

Jumps in sensors can, will and do happen. This is why we make big efforts to write notes and log our sensor systems. Users have to remember Cavet Emptor and use the data wisely and when there are jumps look to reasons, and not mis interpret the data. Us data providers cant hand hold all users. They must do due diligence when using data too. Getting back to my point one should not use all the data. Use what is best and most fitting.

Fig14. Interesting

Discussion

Factors for potential false positive and false negative flagging

Glad to see something on this. But it leaves begging the point I make that respiration pulses are real.

**Detection and interpretation of discontinuities in the time series**

As I have mentioned, these are expected with long term sites as management can make changes..The site history needs to be considered too.

4.3.1 Flagged data points

I have already made my point about the danger of flagging rain pulses that are real. We have studied this with eddy fluxes, chambers, soil probes and they are consistent.

4.3.2 Flagged discontinuities in time series

It is reasonable to flag discontinuities, but aren't they flagged already?

Concluding points

I find this paper on the opaque side. It is a slog to read through, very engineering in spirit, style and narrative.

I must confess given the energy and time to write any paper, this is one I would not have spent writing.

I am missing the 'so what' message and being convinced I need to apply another set of flags to what I am already doing or what is being done in fluxnet, especially something that is automated and may not be applicable for the sites I may need in my synthesis.

The scoring method seems on the arbitrary side and reminds me of the scoring system for the 'best' world universities. Each scoring system yields a different ranking and group. I suspect this would apply to the application of this method, too.

I want to know how often this automated method suffers from type 2 errors, calling an error when there really isn't one.

I want to know how often this automated method suffers from type 2 errors, calling an error when there really isn't one. This concern also revolves around my complaints about flagging real respiration rain pulses. These pulses are real and sustained and should not be flagged (except for the period when the sensors are wet).

At this point I really feel it is up to the editor whether or not they are interested in publishing such a paper. My suspicion is that it may not be cited much, but again I may be wrong. As I look at the data from a different perspective being a data generator and knowing what to belief and accept as reasonable.

---

## Author Response (AR1)

We thank the reviewers for their comments which helped improving the revised manuscript. We first address the major points of the reviewers before we provide a detailed point by point response. For convenience our response is highlighted with green colour.

Major Point I

The main comment shared between both reviews was related to clarity and the way the C2F algorithm was presented, in particular related to the structure of the manuscript. To address this comment we have extensively revised and improved the structure of this part of the paper (section 2). We resolved the previous subsection "Methodological details" by incorporating essential aspects in the relevant previous subsections, and by moving the remaining details to Supplementary Information. For example, (1) we introduced a conceptual diagram on how C2F works as a new Figure 1 (see below), (2) improved Figure 2 (previously Figure 1) to better illustrate the behavior of the outlier score, (3) extended table 2 (definition of constraints) which allowed for moving previous table 3 to SI-1. All these changes were accompanied by extensive and careful revisions of the associated texts to enhance clarity.

[Figure]

**Figure 1: Simplified overview of the C2F approach. a) Definition of consistency constraints with assignments to target variables based on examples for radiation variables. b) Flagging a target variable based on its inconsistency score, which considers multiple indications of inconsistency from several constraints, and based on outliers from single hard constraints. The grey background indicates where a user can modify definitions and settings of C2F. Further steps of the flagging procedure were omitted for clarity here and are described in section 2.2.4.**

Major Point II (by Dennis Baldocchi)

The reviewer pointed out that the framing of the paper in terms of purpose and relevance was not fully clear. To clarify this, we have revised the last paragraph of the introduction to specify the data user community as main audience and to clarify the objectives of the paper:

> "C2F is primarily intended to assist in network wide synthesis studies, e.g. for analyzing the robustness of results to the inclusion of detected data inconsistencies. […] The specific objectives of this paper are to introduce the C2F principles and methodology, and to synthesize detected flux tower data inconsistencies for the widely used FLUXNET 2015 dataset. We illustrate and discuss that patterns of detected flux tower data inconsistencies seem to be associated with issues, which, while generally known in the eddy co-variance community, have not been flagged systematically yet. We provide a critical assessment of the C2F methodology to assist potential users in interpreting the flags, and to guide potential future improvements."

We have inserted statements accordingly in the conclusion section to improve clarity in terms of framing the paper:

> "While the potential existence of such problems is no surprise for eddy covariance specialists, C2F provides associated flags, which were not available before. This is especially useful for synthesis activities, ecosystem modellers, or remote sensing integration with machine learning. We therefore hope that C2F helps in making scientific progress, helps in improving FLUXCOM and process-based models, and helps in flux tower data becoming more accessible and used across communities."

Additionally we make clear it various occasions throughout the manuscript that C2F is complementary and cannot replace procedures and rigor applied by PIs for the raw data. For example in the introduction:

[revised manuscript text omitted]

Reviewer 1 (Dennis Baldocchi)

This is definitely a technical note, and as long as the editors of this journal are willing to consider it, I will give it due diligence as a referee. Otherwise, this document could be used as a white paper or grey literature to supplement processing on the ICOS or Fluxnet web sites.

See major point I and II.

I appreciate the utility of having a set of well defined and accepted flags for data by the community, as data has certain quality due to time and place.  Sadly we are aware that many data users who are not involved in the details and rigors of the measurements, processing and interpretation often ignore these flags.  But no harm in producing and providing them.

We appreciate the positive comment on the general usefulness of flags for eddy covariance data users. We hope that a clear and peer-reviewed documentation raises the awareness of the users and can facilitate a more appropriate usage of the data.

What is unique and distinct here is the production of a complementary set of consistency flags (C2F) for flux tower data, which rely on multiple indications of inconsistency among variables, along with a methodology to detect discontinuities in time series.  I am a fan of multiple constraints, so I am willing to read the case before me.

As I read this work I have some disagreements with conditions which they may flag.

See below for responses on specific conditions.

*For example it is stated that most frequent flags were associated with photosynthesis and respiration during rain pulses under dry and hot conditions.  I have spent a good part of my career studying such pulses.  They are real and sustained following rain events.  To remove them is faulty and will cause biases in sums. Yes, I concur during the rain event itself data such be flagged when sensors are wet. But following the rain, huge amounts of respiration can and will occur.*

See major point III.

***2 Materials and methods***

*As I read this paper back and forth, I wonder about the wisdom of its organization. In the Methods section there are 8 figures or so.  This seems more like a Results and Discussion.  I also suspect much of the excess material could be in an Appendix or Supplemental material.  For a Technical Note, this paper is really long and excessive.*

See major point I. We have extensively and thoroughly revised the Methods section and moved large parts to supplementary material as the reviewer suggested.

*Section 2.2.1.  It is important to benchmark and monitor the relation between PAR and Rg. It is our experience that quantum sensors tend to drift over time, if not frequently calibrated. PAR is used to upscale fluxes with remote sensing and if those relationships are built on faulty values of PAR, the*

*derived products in time, space and trends will be in error. Hence, looking at these flags can have important implications. Too often this issue has been overlooked, so it would be important to know the consequences of improving these data.*

We agree with the reviewer that issues in PAR data could easily propagate or deteriorate downstream analysis, including the flux partitioning. Following the reviewer's comments we have inserted respective statements to acknowledge this aspect more explicitly in the discussion (section 4.3.1):

> "For example quantum sensors to measure photosynthetically active radiation are known to drift over time if not frequently calibrated. C2F could easily be extended to detect this specific problem by assessing trends in the residuals of the SW_IN vs PPFD_IN constraint. […] Clearly, faulty radiation inputs would cause faulty flux predictions."

*The relation between Rn and Rg is important to examine, but realize it will change with season as albedo and surface temperature changes. So be careful and do not make your flags by using one annual dataset for the site.*

We agree with the reviewer on these conceptual issues of the Rn-Rg relationship which we had mentioned explicitly in table 2. For these reasons, the relationship was classified as soft constraint meaning that additional constraints need to indicate outliers for the same data points to cause flagging. The median correlation between daily Rn and Rg is about 0.95 which provides an empirical justification for considering this constraint for deriving inconsistency flags for radiation variables.

*Comparing GPP and Reco with Day vs Night partitioning methods may be interesting, but not sure which is right. We know there is down regulation in dark respiration during the day, it is hard to measure reliable CO2 fluxes at night under stable conditions and with tall vegetation and appreciable storage and or sloping terrain. These can be points of reference and maybe the daily sum is better than hour by hour measurements, as some errors cancel.*

We agree with the reviewer that both, day and night-time partioning methods are complementary and equally useful. Therefore, we provided a parallel assessment of flagging patterns for both variants. To acknowledge the reviewer's comments on challenging factors for measuring CO2 fluxes we inserted a statement in the discussion (4.1.1):

> "This means that C2F is less effective in detecting data issues for NEE and highlights the importance of dedicated checks and corrections applied by PIs especially under challenging conditions of rain, stable atmospheric stratification, sloping terrains, tall canopies, and appreciable storages."

*I have learned from Dario the value of plotting CO2 flux vs u* and developed a matlab subroutine to do so. The threshold can be uncertain as u* has some autocorrelation with the flux. Of course we don't want to set high thresholds as they are based on a diminishing number of data points as high u* values are rare compared to low ones.*

Due to the reason's outlined by the reviewer we used the u* uncertainty quantified according to Pastorello et al. 2020 (that is an extension of Papale et al. 2006 and more robust to noise) as a soft constraint for assessing NEE.

*I see one of the constraints is LE +H vs Rn..What about G or storage in water column? I think these tests are only instructional. We know that there are many differences in sampling areas and representative of radiation and fluxes. It is dangerous to indict one or the other. And with wetlands it is really hard to measure water storage. We have a data set with nearly closed energy balance and then flooded the system and it all went to hell. Same sensors, same processing, ideal fetch and site. Just water moves heat in and out and it is hard as hell to sample well and well enough.*

We agree with the reviewer that accounting for storage can be important for assessing the daily energy balance constraint when e.g. flooding and associated lateral transport of energy would violate the assumption behind the LE+H vs Rn constraint. We have therefore classified this constraint as soft constraint in the revised version and noted this in Table 2:

"Due to the omission of storage changes we classified it as soft constraint."

This implied rerunning C2F and updating all respective Figures in the manuscript while all results remain qualitatively consistent. Overall, the median correlation for this constraint is 0.96 which provides a strong empirical justification for our purpose.

*I must admit I am having a problem coming up with a salient point of this paper and how it will help me do better. I am at the point where an outlier score is proposed. It seems ok, but it is a lot like the college ratings, that depend upon an arbitrary set of metrics and scores.*

See major points I and II.

*I often advise use the set of sites that help you ask and answer the questions you are asking, relating to climate, function and structure. Just because these sites and data are in the fluxnet database does not mean we have to use them all. Maybe this should be the point of this paper.*

Agree. See major point II.

*Figure 1. It is a comparison between machine learning and flux data. Not sure what I am to learn and extract here. Which is right or wrong? Machine learning ultimately is a fancy least squared fit to a bunch of transfer and nodes.*

We improved this figure in the revised version (now Figure 2, see below) to illustrate how the outlier score works and behaves, in particular in the context of heteroscedasticity.

[Figure]

Figure 2: Illustration of the derivation of the outlier score for a constraint. This example is for the machine learning constraint for GPP_NT for US-Wkg. Observed and predicted values are used to calculate residuals and how the distribution of residuals varies with the predicted value to account for

heteroscedasticity. The outlier score on colour and bottom panel measures the distance of the residuals to the quartiles in units of interquartile range (nIQR).

*Here is a set of data comparing annual carbon fluxes with machine learning methods from my sites. They are almost indistinguishable from the direct flux measurements they are derived from. In this case we know our site and develop the machine learning model with the most appropriate and representative biophysical forcings. In the figure given in this paper, I have no idea how appropriate the machine learning model may be for this situation, as the answer is based on independent variables they chose to use or omit.*

The median correlation between observed and machine learning based predictions (cross-validated within site) between 0.93 and 0.99 depending on the target variable (Table 2), which suggests that the chosen predictor variables listed in Table 3 are appropriate. We had invested energy into designing and calculating a set of water availability metrics for improved modelling of water stress effects that are typically more difficult to get (SI-3). The predictor variables were also chosen in context of other C2F constraints to maximize independence among constraints. We improved clarity of this aspect by noting these points early on in the methods section (2.2.1):

> "For the machine learning constraints, the predictions are based on a 3-fold cross-validation with Random Forests (Breimann, 2000) - the target variable specific predictors (Table 2) exclude variables that are already involved in other constraints for the same target variable to maximize independence among constraints. Soil moisture indicator variables were derived from measured precipitation and evapotranspiration (SI-3) and added as predictors to improve the predictability of fluxes under dry conditions."

*Regarding the comparison of radiometers I know during some seasons our guy wires may shade the quantum sensor for certain angles of the sun. surely those data are not fit and I hope such a method may help detect these biases and errors.*

We incorporated the reviewer's suggesting in the context of applying C2F to hourly data in the future (discussion, section 4.2.1):

> "The approach for daily data outlined here can in principle be adapted to sub-daily data but it would require modifying some of the constraints and settings. Hourly C2F could also help in detecting inconsistencies of radiation data for certain sun-angles when parts of the tower infrastructure like guy wires may shade individual sensors."

*Figure 2 seems to be a nice case study to show the attributes of your ideas. Maybe start with that one first. It is clear and more understandable, as we know PAR and Rg are closely related. So when there are differences it can help us think about why and which is more plausible and better.*

In the context of redesigning and revising the Methods section we introduced a new overview diagram as Figure 1 based on the example of radiation variables as the reviewer suggested.

*Fig 3. Maybe I am just tired, or thick, but I don't follow the logic and rationale of the flag for light used for GPP. It would only give me pause on the accuracy of the machine learning calculations, but not the eddy fluxes.*

We clarified that the propagation of flags from SW_IN to GPP_DT is because SW_IN is an input of the daytime based flux partitioning method to fit a light-response curve (section 2.2.4):

> "We propagate flagged data points to dependent variables (e.g. SW_IN is used to calculate GPP_DT during flux partitioning, see table 4 for considered dependencies)"

*Fig 5. I am trying to get my head around the issue of the comparison of the daytime vs nighttime methods. Again, I would argue one is better than the other. Personally, I like the idea of multiple constraints and see if the two methods are converging for confidence, more than anything. Not sure what you all are doing, but in early days working with Eva Falge, we estimated respiration during the day by the extrapolation of the CO2 flux vs light response curve. Now one of the limits is basing a regression and extrapolation on only a few points when the response function is linear, and the fact that during the sunrise sunset period steady state conditions don't hold.  It is these reasons why I argue against one being better or worse, but if they both converge at least we may assume the fluxes may be good enough.*

We clarified that the main intention of Figure 5 is related to illustrate how the flagging behaves for carbon fluxes, and to show that intermediate diagnostics of C2F might be interesting to inspect for EC experts to infer reasons for flagging (section 2.2.4):

> "To further illustrate how multiple indications of inconsistency as well as outliers from hard constraints shape the flagging of carbon fluxes, we look again at the dry site from the US (Fig. 5). […] The examples above illustrate that we can diagnose which constraints have contributed to or caused flagging by inspecting intermediate diagnostics of C2F, which might be interesting for eddy-covariance experts to infer reasons for potential issues in the data."

*The reality is that pulses due to rain or insects passing through the path of the IRGA or sonic are problematic. Or those from electrical noise (a rarity today).  We also see problems with CO2 fluxes over open water as there is a covariance with w and RSSI of the sensor that yields fluxes in the wrong direction and that are not physical.  Those should be filtered. But I don't hear about that here.*

As clarified in detail for major point II, C2F is complementary to and relies on through checking, correcting, flagging done by PIs on the raw data. For example we note in the discussion (section 4.1.1):

> "This means that C2F is less effective in detecting data issues for NEE and highlights the importance of dedicated checks and corrections applied by PIs especially under challenging conditions of rain, stable atmospheric stratification, sloping terrains, tall canopies, and appreciable storages."

*Fig 6 seems to align with my suggestions that some sites may not be the best for some analyses and just toss them. Nothing lost as we oversample in many situations.*

Agree.

*Fig 7. Curious as to why there is a systematic jump in LE.  Eddy covariance should be immune from just a jump as we are doing mean removal. So even if sensors change and they are properly calibrated we should not expect such a marked difference.  This is not like comparing two separate sensors, that can have offsets.*

According to the BADM the break coincides with a major change in instrumentation including a change in the gas analyzer (see also Fig.12). It is an example where these tests could also help the PI to identify at least a missing communication in terms of metadata about the setup.

*Fig 8. Illustration of the outlier score. This is needed to support the method described here. Has taken a long time to get to this point.  Line 350!*

As we agree with the reviewer here we have extensively revised the methods section and moved the respective aspects up to section 2.2.2. See major also point I.

*Results*

*Fig9 demonstrates the point of this method. As expected met variable values tend to have few outliers.*

This is mostly correct while radiation variables show comparatively frequent flagging too.

*Fig 10 provides a needed diagnostic as to when data may be rejected*

Figure 9 (previously Fig.10) is intended to illustrate issues of currently existing flux-partitioning methods in particular for dry and rain pulse conditions, which we hope has been clarified in the revised version (section 3.1):

> "These conditions correspond typically to very dry conditions, where the assumptions of the NEE flux partitioning methods are more frequently violated: ecosystem respiration is less controlled by temperature, and GPP is less limited by light. Visual inspection of the time series (e.g. Fig. 5) suggested particular flux partitioning issues during respiration rain pulses, where e.g. GPP_NT is often systematically negative while NEE is elevated. We found a systematic pattern of strongly elevated flagging frequency during and after rain when temperatures are high (>15°C) and GPP is low (Figure 9). This illustrates methodological limitations of the flux partitioning methods in dealing with rapid changes of ecosystem responses due to the used moving window approach to estimate parameters during flux partitioning processing."

*Fig 11.  Would think this would be a function of open vs closed path
sensors*

Thank you for this interesting hypothesis which we consider for a follow-up study.

*Fig 13. The jumps in NEE seem to be with site management.  So Know Thy Site.  Just don't blindly process long term data. This is why we have phenocams at our tower, to look at the vegetation when things are 'weird'.*

Agree. Management is clearly an important factor that can cause flags for temporal discontinuities as shown in figure 12, described in section 3.2, and discussed in section 4.1.2. Changes in instrumentation or processing by the PI seem to be another important, and apparently the more important cause of temporal discontinuities (see Figure 13 and its caption and section 4.1.2). As discussed in section 4.2.1

the purpose of these flags is to make users aware of temporal discontinuities and then the decision of how to interpret or deal with them is left to the user according to the requirements:

> "This implies that detected discontinuities require careful attention in order to judge whether it is due to an artefact or a real phenomenon and it confirms the importance of complete metadata and ancillary data as crucial set of information for the proper interpretation of the measurements. […] How to deal with detected discontinuities in the time series can also be very application dependent and may vary from discarding the site, keeping only the longest segment, running the analysis separately within segments, or not doing anything about it. Clearly, analysis targeting interannual variations or trends should consider discontinuities in the time series that could be artefacts of changes in the measurement setup."

*Jumps in sensors can, will and do happen. This is why we make big efforts to write notes and log our sensor systems.  Users have to remember Cavet Emptor and use the data wisely and when there are jumps look to reasons, and not mis interpret the data.  Us data providers cant hand hold all users.  They must do due diligence when using data too. Getting back to my point one should not use all the data.  Use what is best and most fitting.*

While we agree in principle we think that the community should try its best to facilitate an appropriate usage of FLUXNET data for non-EC experts (in particular when large number of sites are used in an automatic ingestion system) to ultimately facilitate a wide and solid use of the data. In section 3.2 and Figure 12 we show that 1) jumps can be associated with changes in instrumentation, while often they are probably not (which is good), and 2) jumps are likely associated with changes in instrumentation that was not reported – respective meta data in the BADMs is very heterogeneous with respect to quality and completeness between sites, and non-trivial to use for non EC experts.

Since we are addressing this problem explicitly we hope our flags will be complementary and useful for the community of data users. A conceptual advantage of our system compared to visual expert judgement is that the approach is standardized and automated, avoids subjectivity and avoids potential confirmation bias in selecting or filtering out data. See also major point 2

*Fig14. Interesting*

Thank you.

*Discussion*

*Factors for potential false positive and false negative flagging*

*Glad to see something on this. But it leaves begging the point I make that respiration pulses are real.*

See major point III.

*Detection and interpretation of discontinuities in the time series*

*As I have mentioned, these are expected with long term sites as management can make changes..The site history needs to be considered too.*

Agreed. The flags can help pointing to management effects on fluxes as discussed in section 4.1.2 and 4.2.1, and the user can consult the BADMs and the PI to find out what happened specifically. Likewise, changes in instrumentation seem to play another major role unfortunately. This demonstrates once more the importance of the management and instrumentation change data that are often not shared, which we have emphasized in the discussion (section 4.3):

> "This calls again to the importance of the maintenance of the sensors and the correct and full recording and reporting of all sensors replacements or calibrations in the metadata."

> "Also, in this case the availability of metadata about sensors or setup change or major disturbances/management activities at the sites are very important for the interpretation of detected discontinuities and could allow for more tailored approaches in the future."

*4.3.1 Flagged data points*

*I have already made my point about the danger of flagging rain pulses that are real.  We have studied this with eddy fluxes, chambers, soil probes and they are consistent.*

See major point III.

*4.3.2 Flagged discontinuities in time series*

*It is reasonable to flag discontinuities, but aren't they flagged already?*

No, they are not. Therefore we think our methodology and tool will be useful.

*Concluding points*

*I find this paper on the opaque side.  It is a slog to read through, very engineering in spirit, style and narrative.*

See major point I and II. We thank the reviewer for his critique that helped improving the revised version of the manuscript.

*I must confess given the energy and time to write any paper, this is one I would not have spent writing.*

See major point II.

*I am missing the 'so what' message and being convinced I need to apply another set of flags to what I am already doing or what is being done in fluxnet, especially something that is automated and may not be applicable for the sites I may need in my synthesis.*

See major point II.

*The scoring method seems on the arbitrary side and reminds me of the scoring system for the 'best' world universities. Each scoring system yields a different ranking and group. I suspect this would apply to the application of this method, too.*

Since we aim at flagging inconsistencies in flux tower data based on a clear rationale and a set of conceptually and empirically justified criteria we think the analogy to college ratings does not apply here. See also major point II.

*I want to know how often this automated method suffers from type 2 errors, calling an error when there really isn't one.*

We fully agree with the reviewer that this would be relevant to know. To facilitate such analysis we either need labels for the real data or synthetic data. We would love to have them but don't. It is clearly a function of the chosen nIQR threshold (see discussion in section 4.1.1) – when increasing nIQR (allowing for more loose consistency) type 2 errors will decrease:

"Increasing nIQR makes C2F becoming more loose leading to less flagged data overall, less false positives but more false negatives."

In the conclusion section, we encouraged that future efforts should try to establish such a benchmarking data set such that we can objectively evaluate type 1 and type 2 errors and ultimately improve the method:

"To further develop and improve C2F it would be desirable to be able to benchmark it objectively using a large set of synthetic data, where flux tower data with all its potential issues and noise properties are realistically emulated with labels for inappropriate data available."

*This concern also revolves around my complaints about flagging real respiration rain pulses. These pulses are real and sustained and should not be flagged (except for the period when the sensors are wet).*

See major point III.

*At this point I really feel it is up to the editor whether or not they are interested in publishing such a paper. My suspicion is that it may not be cited much, but again I may be wrong. As I look at the data from a different perspective being a data generator and knowing what to belief and accept as reasonable.*

See major point II. We thank the editor and the anonymous reviewer for seeing and appreciating the value of C2F for the scientific community. We hope with our response, revisions, and clarifications we could also convince this reviewer.

Reviewer 2

GENERAL COMMENTS

In this manuscript a new algorithm for quality flagging eddy covariance (EC) flux data is proposed. Long EC flux time series are already available from several measurement sites and the FLUXNET datasets consists of observations from hundreds of sites. Such datasets are an invaluable source of information for studies focusing on land-atmosphere interactions. However, there can be spurious differences between sites (e.g. due to instrumentation or different data processing pipelines) and spurious temporal discontinuities in long time series which complicate the usage of this data. Such problems have been minimized in research infrastructures such as ICOS and NEON, where instrumentation and data processing have been standardized and sufficient metadata are available. However, for older data and data stemming outside these standardized infrastructures these problems may still persist. This manuscript tries to find a solution to these problems with additional quality flagging of EC flux time series.

The manuscript is within the scope of BG (although might fit better to AMT or GI due to its technical nature) and presents novel ideas for solving an existing problem (which would not however exist if all the needed metadata would be available). The scientific quality of the work is good, but to a large extent the presentation quality is not. My main criticism is directed towards how the new algorithm is presented, in particular towards Sect. 2 in the manuscript. The section is very difficult to follow, and the reader needs to constantly jump back and forth between subsections when reading the text. For instance, when I reached Sect. 2.4 I realized that I had read the whole section already, since I needed to read it simultaneously with the sections 2.2. and 2.3 in order to understand the text. It took me quite long time to understand how the whole algorithm works. Hence, I strongly suggest rethinking the structure of the text in Sect. 2.

We thank the reviewer for this assessment which helped improving the quality of the manuscript. We have substantially revised the methods section as the reviewer suggested as detailed in our response to major point I.

I suggest accepting this manuscript after major revisions, mainly due to the way the algorithm is presented in the manuscript.

As a sidenote, this is one of those manuscripts where it would be really helpful for the reviewer if the underlying code and a small example dataset would be available already during the review. However, currently it does not seem to be prerequisite for manuscript submission in this journal and hence do not expect the authors to make such material available at this stage.

The code together with example data are made available with the revised version.

SPECIFIC COMMENTS

Row 40: "spectral corrections" plus other processing steps, e.g. coordinate rotation.

Agree. We accommodated this accordingly.

Rows 75-78: I suggest that you add references for these research questions. They are clearly related to prior work. Now it reads like that the reader should already know that e.g. interannual variability of sensible heat flux can be predicted better than interannual variability of latent heat flux or that the reader knows what is "the issue to model drought effects in GPP".

Agree. We added context and references here. It now reads:

"For example, from the perspective of machine learning based flux modelling by the FLUXCOM approach (Jung et al. (2019), Jung et al. (2020)), some unanswered example questions on the contribution of potential flux tower data issues include (Bodesheim et al. (2018), Tramontana et al. (2016)): (1) Can we predict the interannual variability of sensible heat flux much better than that of latent heat flux due to differential observational uncertainties? (2) To what extent is the low skill in predicting NEE interannual variability at FLUXNET site level due to temporal discontinuities arising from changes in instrumentation and set-up. (3) How much of the issue to model drought effects in GPP is due to flux partitioning problems? (4) Where is the optimal trade-off between data quantity and data quality used for training machine learning models?"

Row 86: FLUXCOM not introduced anywhere, you need to briefly tell what it is.

Agree. We followed the suggestion – see our response to the previous comment.

Rows 102-102: I have not used FLUXNET2015 data, is it originally with daily time step or did you average it to daily values? Please mention in the text.

Sorry for this missing information. The original data are (typically) half-hourly – this is now clarified:

"The FLUXNET2015 Dataset (Pastorello et al., 2020) is a collection of half-hourly meteorological and flux data measured at 212 sites and collected from multiple regional flux networks."

Rows 102-103: What is fqcOK, a variable in FLUXNET2015 dataset? Consider removing it from the text and just write that you removed those days from the analysis for which more than 20 % of data were not measured or gapfilled with high confidence.

Agreed and changed:

"We keep only daily data points that are based on at least 80% of measured data or gap-filled with high confidence."

Row 112: "expected relationship", this can be dangerous as you are enforcing a certain dependence between variables. By doing this you may inadvertently quality flag (and screen out) scientifically interesting periods. This ought to be discussed this in the text.

We apologize that our formulation was a bit misleading because the detection of an outlier from an expected relationship is only one indication for inconsistency, while the inconsistency score and the flagging consider multiple independent indications of inconsistency. We think our extensive revisions of the methods section (see major point I) clarifies this aspect.

Row 159: I suggest adding a site ID in parentheses after "United States".

Done

Row 161: what is F15? Please clarify

F15 stands for the FLUXNET 2015 data set. We avoided the abbreviation in the revised version.

Row 319: "This made the outlier score too sensitive to very small residuals even." What does this mean? Please clarify

We thank the review for pointing out insufficient clarity. Basically the problem is that during the calculation of the outlier score we divide by the interquartile range of the residuals, which can be (close to ) zero in some occasional cases. In these rare cases of extremely small interquartile range of residuals the outlier score could become very large even for a tiny absolute residual. We clarified this accordingly in the revised version (SI-2):

> "In these rare cases of extremely small interquartile range of residuals the outlier score could become very large even for a tiny absolute residual due to dividing by a number close to zero."

Row 349: Does the machine learning (ML) model predictive performance have an impact on these ML constraints? Do you assume that the model residuals are only related to noise in the measurements, i.e. the model is perfect? I suggest discussing this briefly in the text, e.g. here, in Sect. 2.4.8 or other suitable location. ML models typically perform worse for NEE than e.g. for GPP (see e.g. Tramontana et al., 2016) and hence this constraint might not work similarly for all variables.

These are all important considerations that were incorporated in the design of the method. We explicitly assume ML models to be imperfect models with underlying assumptions – for this reason the ML models are classified as soft constraints as mentioned in Table 2. While the reviewer correctly pointed out that predicting NEE is harder than other fluxes, we show in Table 2 that the median correlation with the cross-validated ML predictions is above 0.93 for all variables. This is because the main difficulty of predicting fluxes is between sites, not within a given site. We further clarify in section 2.2.2 that the outlier scores are comparable among each other even though they may differ in terms of performance:

> "Outlier scores from different constraints are independent of units, comparable, and therefore combinable among different constraints, which is an important prerequisite to calculate the inconsistency score later. Accordingly, this facilitates combining outlier scores from different constraints with different empirical strength of the relationships because the outlier score for a constraint is relative to the spread of the residuals."

Row 357: you need to introduce scikit-learn

Done.

Rows 373-374: "Six variants of tCWDt C with different C values of 15,50,100,150,200,250 mm were calculated." Already mentioned above on rows 368-369. Please remove

Done.

Row 377 (Table 5): These variable names most likely follow FLUXNET2015, but you need to tell the reader what these variables are. Currently, they are not all introduced in the text.

We expanded table 1 to introduce these ancillary flux tower variables.

Row 604: I would argue that false negative is not as bad as false positive (conservative approach).

In general, we agree with the reviewer and in fact the design of the methodology tried to minimize false positives by introducing soft vs hard constraints, by requiring 2 or more indications of inconsistency from independent constraints, by accounting for heteroscedasticity, and by choosing a default value of nIQR=3. All these points are discussed in section 4.1.1. However, there are certainly also potential applications where a very strict data filtering may be desired and we provide this flexibility here by allowing to vary the nIQR strictness parameter to accommodate this (discussion, section 4.1.1):

> "By varying the nIQR parameter we can choose how strictly we apply C2F as this determines how far into the tails of the distribution of residuals a data point is allowed to fall. Increasing nIQR makes C2F becoming more loose leading to less flagged data overall, less false positives but more false negatives."

TECHNICAL CORRECTIONS

We thank the reviewer for the thorough check of our manuscript and for providing these technical corrections which we implemented all in the revised version of the manuscript.

Row 52: extra ")" after "Drought2018"

corrected

Row 77: replace "." with "?"

corrected

Row 87 and maybe elsewhere: you use both ONEFLUX and ONEFlux. Use only one of these two, check Pastorello et al. (2020).

We now use ONEFlux consistently as introduced by Pastorello et al.( 2020)

Row 95: replace "FLUXNET" with "FLUXNET2015"

Done

Row 102: replace "table" with "Table".

Done

Row 115: should "2.2.2" be replaced with "2.2.1"?

This paragraph was substantially revised and we checked carefully for referencing the correct sections.

Row 232: replace "Turing" with "Turning"

corrected

Row 592 (Figure 14): Colorbar label is incomplete in the right plot

corrected